# Probing How Scalable Table Data Enhances General Long-Context Reasoning

**Huaibing Xie** [* 1]  **Guoliang Zhao** [* 1 2]  **Yang Liu** [* 1]  **Shihan Dou** [1]  **Siming Huang** [3]  **Yanling Xiao** [1]  **Shaolei Wang** [1]
**Yiting Liu** [1]  **Cheng Zhang** [1]  **Shaofan Liu** [1]  **Pluto Zhou** [1]

## Abstract

As real-world tasks grow increasingly complex, long-context reasoning has become a core capability for Large Language Models (LLMs). However, few studies explore which data types are effective for long-context reasoning and why. We find that structured table data with periodic structures shows strong potential for long-context reasoning. Motivated by this observation, we mathematically analyze tabular dependency structures using mutual information, revealing periodic non-vanishing dependencies in table data. Furthermore, we systematically analyze the capabilities of structured table data, conduct relevant scaling experiments, and validate its underlying mechanisms for enhancing long-context reasoning, yielding several meaningful insights. Leveraging these insights, we propose a simple yet scalable pipeline(TableLong) for synthesizing high-quality, diverse, and verifiable structured table data to boost long-context reasoning via RL. Extensive experimental results demonstrate that table data significantly enhances the long-context reasoning capability of LLMs across multiple long-context benchmarks (+8.24% on average), and even improves performance on out-of-domain benchmarks (+8.06% on average). We hope that our insights provide practical guidance for effective post-training data to enhance long-context reasoning in LLMs.

## 1. Introduction

As real-world tasks become increasingly complex, long-context reasoning has emerged as a fundamental capability of Large Language Models (LLMs) (Li et al., 2024b). Reinforcement Learning (RL) for long-context reasoning requires LLMs to retrieve and ground key information from long-context inputs, and then generate Chain-of-Thought (CoT) reasoning based on the integrated information (Guu et al., 2020; Ram et al., 2023; Wan et al., 2025; Shen et al., 2025). Existing studies on long-context reasoning in LLMs primarily focus on training strategies and methodologies (Yang et al., 2025a; Team et al., 2025; Comanici et al., 2025; Zeng et al., 2025; Wan et al., 2025; Shen et al., 2025).In addition, some emerging works have started to focus on synthetic data pipelines for long-context reasoning tasks (Zhu et al., 2025b; Yang et al., 2025b; Zhang et al., 2025). Furthermore, table data has also been introduced as a benchmark to evaluate long-context capabilities (Wang et al., 2025a). However, few studies have analyzed how the intrinsic characteristics of different types of data influence the long-context reasoning capability of LLMs, which holds significant implications for effective post-training data to enhance long-context reasoning in LLMs.

Ideal long-context reasoning data should possess long-range dependencies, scalability, and verifiability. However, natural text, and even some synthetic plain text, often fail to satisfy all of these abilities simultaneously. In contrast, table data with periodic non-vanishing structural property demonstrates strong scalability, verifiability, multi-hop reasoning ability, and knowledge grounding ability, which exhibits strong potential for adapting to long-context reasoning capability in LLMs (Wu et al., 2025; Lei et al., 2025). Therefore, our work aims to investigate the impact of structured table data on the long-context reasoning capability of LLMs and to dissect the underlying mechanisms driving this effect.

To achieve the aforementioned objectives, we first mathematically analyze tabular dependency structures via mutual information, revealing the existence of periodic non-vanishing dependencies. We then validate the mechanisms by which table data enhances long-context reasoning through systematic scaling experiments. Leveraging these insights, we propose a simple yet scalable pipeline(TableLong) for synthesizing high-quality, diverse, and verifiable structured table data to boost long-context reasoning via RL. Specifically, we synthesize diverse SQL tasks with verifiable execution results, applying a consistency-

---

[*]Equal contribution  [1]Large Language Model Department, Tencent  [2]Xi'an Jiaotong University, Xi'an, China  [3]Fudan University, Shanghai, China.  Correspondence to: Huaibing Xie <huaibingxie@tencent.com>, Guoliang Zhao <zgl934455716@stu.xjtu.edu.cn>.

*Proceedings of the 43ʳᵈ International Conference on Machine Learning*, Seoul, South Korea. PMLR 306, 2026. Copyright 2026 by the author(s).

based filter to prune noise and retain challenging samples. By incorporating structured table data into RL training, we observe significant performance improvements across the majority of long-context benchmarks, and further validate its generalization capability on Out-of-Domain (OOD) benchmarks, such as Math (AIME 2025)(Balunović et al., 2025) and Code (LiveCodeBench)(Jain et al., 2024). Overall, our key contributions are as follows:

1. We are the first to mathematically analyze tabular dependency using mutual information, revealing periodic non-vanishing structural property of table data.

2. We propose a simple yet scalable pipeline for synthesizing high-quality, diverse, and verifiable structured table data tailored for long-context reasoning.

3. Results demonstrate that table data significantly enhances the long-context reasoning capability of LLMs, generalizes to other domains, and we systematically decompose and analyze it to derive meaningful insights.

## 2. Mutual Information Analysis of Table Data

Table data is widely used in real-world applications, yet its structural properties for LLM training remain underexplored. Unlike natural language with sequential tokens and decaying contextual relevance, table data exhibits rigid column-wise structures that induce distinct dependency patterns.

We provide an information-theoretic framework to analyze long-context dependencies in table data. We show that tables preserve **periodic non-vanishing** structural dependencies, in contrast to the power-law decay observed in natural language. To our knowledge, this is the first mutual information-based analysis of tabular dependency structures. These properties make table data well suited for training long-context reasoning capability of LLMs.

### 2.1. Background: Dependency Decay in Natural Language

For natural language sequences, empirical studies (Li, 1989; Lin & Tegmark, 2017) demonstrate that Mutual Information (MI) between tokens decays polynomially with distance ($d$):

$$I_{text}(d) \sim C \cdot d^{-\alpha}, \quad \text{with } \alpha \approx 0.5 \quad (1)$$

This power-law decay, which we adopt as a modeling assumption for natural language throughout this work, implies $\lim_{d \to \infty} I_{text}(d) = 0$. A natural question arises: *does table data exhibit the same decay pattern?*

### 2.2. Setup: Table Data and Linearization

A table $\mathcal{T}$ is an $n \times m$ structure with column headers $\mathcal{H} = \{H_1, \ldots, H_m\}$ and cells $\{T_{i,j}\}_{i \in [n], j \in [m]}$, where $[n] := \{1, \ldots, n\}$ and each cell $T_{i,j}$ is drawn from a column-specific distribution $P_j$. The row-major linearization $\phi : \mathcal{T} \to W_{1:L}$ maps a table to a token sequence $\phi(\mathcal{T})$:

$$[H_1, \ldots, H_m, \underbrace{T_{1,1}, \ldots, T_{1,m}}_{\text{row 1}}, \underbrace{T_{2,1}, \ldots, T_{2,m}}_{\text{row 2}}, \ldots, T_{n,m}]$$

with total length $L = m(n+1)$, where $W_t$ denotes the $t$-th token. For position $t > m$ in the data region, let $\text{col}(t) \in [m]$ denote its column index.

For lag $d \in \{1, \ldots, L - m - 1\}$, define the average MI over the data region:

$$\bar{I}_{table}(d) = \frac{1}{|\mathcal{S}_d|} \sum_{t \in \mathcal{S}_d} I(W_t; W_{t+d}),$$

$$\mathcal{S}_d = \{t : m < t \leq L - d\}$$

Our analysis rests on the following assumptions (formal statements in Appendix A.2):

(A1) *Column Semantic Consistency*: All cells in column $j$ share the same distribution $P_j$.

(A2) *Column Distribution Distinctiveness*: Different columns have distinguishable distributions, i.e., $D_{KL}(P_j \| P_k) > 0$ for $j \neq k$, where $D_{KL}$ denotes the Kullback-Leibler divergence.

We do **not** assume that cells within the same column are conditionally independent; our results hold regardless of intra-column dependencies (see Remark A.7 in the Appendix).

### 2.3. Periodic Non-Vanishing Dependencies

We first introduce a key factor. Let $J \sim \text{Uniform}([m])$ be a randomly chosen column index.

**Definition 2.1** (Same-Column Mutual Information). The same-column MI is defined as:

$$I^{same} := I(T_{i,J}; T_{k,J}) \quad (2)$$

As shown in Lemma A.8, under Assumptions (A1)–(A2): (i) $I^{same} > 0$, and (ii) $I(T_{i,J}; T_{k,J}) = I^{same}$ for any $i \neq k$, i.e., same-column MI is independent of row distance.

The magnitude of $I^{same}$ depends on how distinguishable the column distributions are. As characterized in Proposition A.11, $I^{same}$ is determined by the cross-column variance $\sum_a \text{Var}_j[P_j(a)]$, where $\text{Var}_j[\cdot]$ denotes variance across columns. When column distributions are highly distinct, this variance is large, yielding larger $I^{same}$.

**Theorem 2.2** (Periodic Non-Vanishing Dependency). *Under Assumptions (A1)–(A2), for any table with $n$ rows and $m$ columns:*

$$\bar{I}_{table}(km) = I^{same} > 0, \quad \forall k \in \{1, 2, \ldots, n-1\} \quad (3)$$

*That is, $\bar{I}_{table}(d)$ attains periodic peaks of constant height $I^{same}$ at every multiple of the column count $m$.*

Note that non-vanishing property holds *specifically at periodic lags $d = km$; at others, average MI may be smaller.*

**Corollary 2.3** (Asymptotic Non-Decay). *For a sequence of tables with $n \to \infty$ rows and fixed $m$ columns, under Assumptions (A1)–(A2):*

$$\liminf_{d \to \infty} \bar{I}_{table}(d) \geq I^{same} > 0 \qquad (4)$$

*This contrasts starkly with natural language, where $\lim_{d\to\infty} I_{text}(d) = 0$ based on Eq. (1).*

**Corollary 2.4** (Asymptotic Dominance over Natural Language). *Under the power-law assumption Eq. (1) for natural language and Assumptions (A1)–(A2) for tables:*

$$\lim_{k \to \infty} \frac{\bar{I}_{table}(km)}{I_{text}(km)} = +\infty \qquad (5)$$

### 2.4. Effective Dependency Distance

**Definition 2.5** (Effective Dependency Distance (Liu, 2008)). Given threshold $\tau > 0$: $D_{eff}(\tau) = \sup\{d : I(d) \geq \tau\}$.

**Theorem 2.6** (Effective Distance Comparison). *Under the power-law assumption Eq. (1) for natural language and Assumptions (A1)–(A2) for tables, for any $\tau \in (0, I^{same})$:*

$$D_{eff}^{table}(\tau) = +\infty, \quad while \quad D_{eff}^{text}(\tau) < +\infty \qquad (6)$$

This result shows that for any threshold $\tau < I^{same}$, tabular dependencies exceed $\tau$ at infinitely many lags (specifically, at all $d = km$), whereas natural language dependencies eventually fall below $\tau$ permanently.

These results indicate that structured table data with periodic non-vanishing structures exhibits strong potential for long-context reasoning.

## 3. TableLong: a Scalable and Verifiable Pipeline for Long-context Reasoning

In this section, we propose a simple yet scalable pipeline, namely TableLong, to construct high-quality, diverse, and verifiable structured table data, tailored for RL.

### 3.1. Overview of the Construction Pipeline

As illustrated in Figure 1, our framework achieves a pipeline to transform hybrid tabular sources into high-quality RL tasks. The pipeline proceeds in three stages:

- **Environment Initialization:** We first aggregate a diverse repository of tables from both open-source datasets and documents, parsing them into executable SQL environments.

- **Sample Construction:** Leveraging these environments, we employ LLMs to generate diverse SQL queries and corresponding natural language questions, and subsequently execute the SQL queries to obtain verifiable answers. We construct each instance as a tuple: (Raw Table, Question, Answer).

- **Verification and Filtration:** We then apply a consistency-based filtration mechanism to prune noise ($P = 0$) and triviality ($P = 1$), ensuring the selected tasks offer optimal training value.

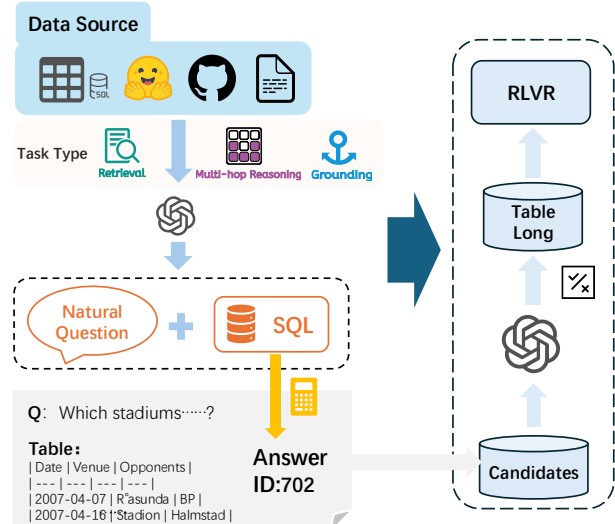

*Figure 1.* Overview of TableLong: An end-to-end table data construction pipeline for long-context reasoning.

### 3.2. Environment Setup

To ensure the diversity and realism of the reasoning environment, we construct a hybrid repository by aggregating over 10,000 real-world tables from established datasets (e.g., BIRD (Li et al., 2023), CoSQL (Yu et al., 2019), Spider (Yu et al., 2018)) and extracting scalable table data from information-dense documents via LLMs.

Following rigorous cleaning and sampling, the resulting corpus exhibits high diversity in both content and structure. Thematically, the data spans a broad spectrum of domains, including Finance, Sports, Healthcare, and Science. Linguistically, it covers both English and Chinese. Crucially, the corpus is tailored for long-context reasoning: context lengths range from a few hundred up to 32k tokens, with an average cell density of 5.2 tokens. Furthermore, to support complex structural grounding, the number of tables per instance ranges from single to 30 tables, creating a challenging multi-table environment. Finally, all collected tables are parsed into a unified SQLite database to support executable SQL queries.

## 3.3. Sample Construction

We generate diverse SQL queries by scalable prompt constraints, targeting **three distinct dimensions** of table tasks for long-context reasoning:

- **Precise Retrieval:** Targeting atomic information locating. We constrain the generation to `SELECT` queries with specific `WHERE` clauses, requiring the model to filter and extract precise values from contexts.

- **Multi-hop Reasoning:** Targeting long-range multi-hop reasoning. We constrain the generation to column-wise aggregation and calculation (e.g., `SUM`, `AVG`), requiring the model to attend to non-adjacent tokens separated by substantial distances due to table linearization.

- **Grounding:** Targeting query-to-table association. We constrain the generation to multi-table operations (e.g., `JOIN`), requiring the model to distinguish specific tables distributed across the long context as "grounding".

Upon generating diverse SQL queries and corresponding natural language questions, we execute the queries to obtain verifiable answers. Finally, we construct each training sample as a triplet: (Raw Table, Question, Answer).

## 3.4. Verification and Filtration

To ensure appropriate task difficulty, we implement a consistency-based filtration mechanism. For each instance $(x, y_{gt})$, we generate $N$ candidate responses using the model and evaluate their correctness against the ground truth $y_{gt}$. Based on the calculated pass rate $(P)$, we apply a **dual-sided filter**:

- **Discard** $P = 0$**:** Eliminates tasks that are ambiguous, erroneous, or exceed reasoning capacity.

- **Discard** $P = 1$**:** Prunes trivial tasks that the model has already mastered, ensuring high training efficiency.

- **Retain** $0 < P < 1$**:** Preserves non-trivial tasks within the effective learning boundary, offering optimal gradient signals for RL training.

# 4. Experiments

## 4.1. Setup

**Benchmarks.** To evaluate OOD universality and scalability, we conduct evaluation across diverse long-context tasks. We utilize **LongBench-v2** (Bai et al., 2025) and **Loong** (Wang et al., 2024) for holistic real-world assessment. Furthermore, we include specific diagnostic datasets: **Browscomplong** (OpenAI, 2025a) and **MRCR** (OpenAI, 2025b) for retrieval;

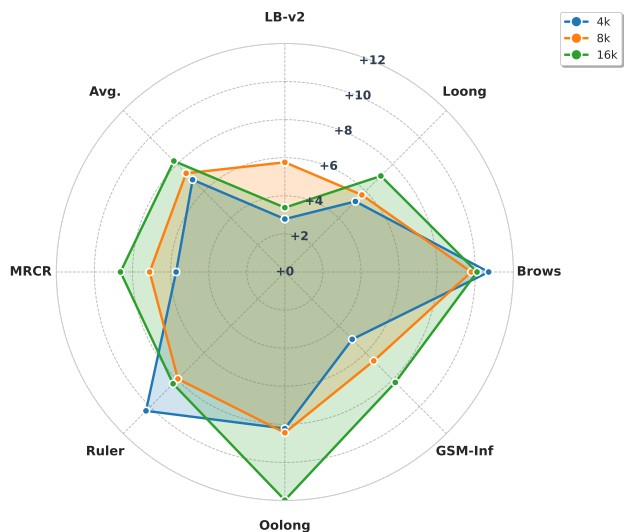

*Figure 2.* The radar chart of long-context reasoning benchmarks for DS-R1-Distill-32B trained with varying length.

the **Ruler** (Hsieh et al., 2024) subset for variable tracking; and **GSM-Infinite** (Zhou et al., 2025) alongside **Oolong** (Bertsch et al., 2025) for numerical reasoning. To assess out-of-distribution (OOD) generalization, we further evaluate on **LiveCodeBench**(Jain et al., 2024), **AIME 2025**(Balunović et al., 2025), **GPQA-Diamond** (Rein et al., 2023), and **MultiChallenge** (Deshpande et al., 2025). Details of benchmarks are provided in Appendix C.

**Baselines.** We take four distinct backbones as our baselines: Qwen2.5-32B-Instruct (Team, 2024), Qwen3-32B (Team, 2025), Deepseek-R1-Distill-Qwen-14B, and Deepseek-R1-Distill-Qwen-32B (DeepSeek-AI, 2025), and apply RL training on them using our TableLong data. Additionally, we include Gemini-3-Pro, Deepseek-v3.1 (DeepSeek-AI, 2024)and Qwen-Long-L1 (Wan et al., 2025) as external baselines. Among them, Qwen-Long-L1, also based on Deepseek-R1-Distill-Qwen-32B and trained for long-context reasoning via RL, serves as a relevant reference for same-backbone performance.

**Training Details.** Experiments are conducted on 64 H20 GPUs using the Verl (Sheng et al., 2024) framework based on the GRPO (Shao et al., 2024) algorithm. Details of training configurations are in Appendix B.

## 4.2. Main Results

In this subsection, we evaluate the effectiveness of our Table-Long in enhancing the general long-context reasoning capabilities of LLMs across multiple OOD general long-context reasoning benchmarks using several open-source backbone models.

*(a) Scalable table data is effective for long-context rea-*

*Table 1.* **Performance Comparison on Long-Context and Reasoning Benchmarks.** We compare base models against their variants optimized via Reinforcement Learning on our dataset (denoted as **+ Ours**). The best results within each model family are highlighted in **bold**. LB-v2: LongBench v2; BROWS: BrowsCompLong; GSM-INF: GSM-Infinite; OOLONG: Oolong-Synth. For model names, Qwen2.5-32B-Inst denotes Qwen2.5-32B-Instruct, and DS-R1-Distill refers to Deepseek-R1-Distill-Qwen.

| Model | LB-v2 | LOONG | BROWS | GSM-INF | OOLONG | RULER | MRCR | Avg. |
|---|---|---|---|---|---|---|---|---|
| *Reference Flagships & Baselines* | | | | | | | | |
| Gemini-3.0-pro | **69.38** | **65.43** | **88.07** | **87.06** | **78.41** | **83.01** | **75.30** | **78.09** |
| Deepeek-v3.1 | 52.88 | 50.55 | 56.27 | 47.40 | 51.90 | 42.97 | 46.62 | 49.80 |
| Qwen-Long-L1 | 43.74 | 44.68 | 69.93 | 9.00 | 27.37 | 47.70 | 27.70 | 38.59 |
| *Main Results: Effectiveness of Our Method* | | | | | | | | |
| Qwen3-32B | 46.33 | 39.96 | 59.33 | 12.22 | 31.66 | 50.86 | 42.45 | 40.40 |
| Qwen3-32B **+ Ours** | **46.92** | **43.10** | **65.44** | **23.40** | **42.31** | **53.55** | **42.66** | **45.34** |
| Qwen2.5-32B-Inst | 38.17 | 33.22 | 52.56 | 9.60 | **41.03** | 45.25 | 33.19 | 36.15 |
| Qwen2.5-32B-Inst **+ Ours** | **43.54** | **38.18** | **60.55** | **13.00** | 35.50 | **62.65** | 33.19 | **40.94** |
| DS-R1-Distill-14B | 35.19 | 25.07 | 51.38 | 9.00 | 42.05 | 61.28 | 29.02 | 36.14 |
| DS-R1-Distill-14B **+ Ours** | **38.97** | **35.44** | **74.31** | **9.60** | **45.95** | **80.72** | **30.48** | **45.07** |
| DS-R1-Distill-32B | 42.35 | 38.17 | 64.22 | 6.60 | 39.43 | 58.11 | 31.94 | 40.12 |
| DS-R1-Distill-32B **+ Ours** | **45.73** | **45.30** | **74.31** | **14.80** | **51.41** | **66.41** | **40.57** | **48.36** |

*(a)* DS-R1-Distill-14B

*(b)* DS-R1-Distill-14B + Ours

*(c)* DS-R1-Distill-32B

*(d)* DS-R1-Distill-32B + Ours

*Figure 3.* Needle in a Haystack retrieval across document depths. Our approach significantly enhances long-context robustness, boosting the 14B model's accuracy from 69.30% to 91.20% and the 32B model's accuracy from 87.95% to 99.40%, achieving near-perfect performance.

*soning.* Specifically, as shown in Table 1, for Deepseek-R1-Distill-Qwen-32B, our TableLong achieves a remarkable improvement, with an average accuracy gain of 8.24% across seven OOD benchmarks of long contexts. Similarly, for other backbones (Qwen3-32B, Qwen2.5-32B-Instruct, and Deepseek-R1-Distill-Qwen-14B), the improvements are 4.94%, 4.79%, and 8.93%, respectively. Deepseek-R1-Distill-Qwen-32B with our TableLong clearly outperforms Qwen-Long-L1 and approaches the performance of Deepseek-v3.1. This demonstrates the robustness of our TableLong in enhancing long-context reasoning capabilities.

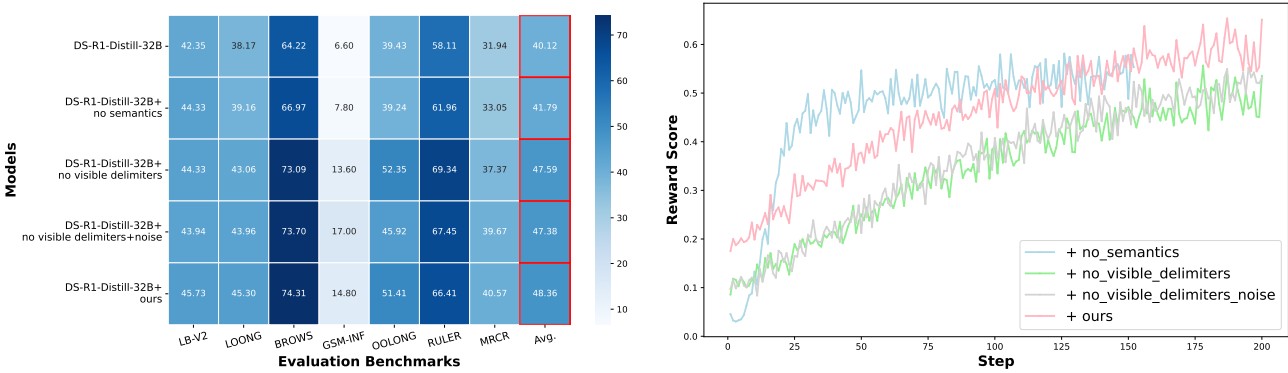

*(a)* Impact of semantics, delimiters, and noise.  *(b)* RL reward curves: Convergence and stability analysis.

*Figure 4.* **Decomposition experiments for DS-R1-Distill-32B.** (a) While structure alone ("no semantics") boosts baseline performance (+1.67%), semantics remain essential for peak results. Removing delimiters or adding **noise** yields negligible drops, confirming the primacy of intrinsic structure. (b) Models with "no semantics" suffer premature convergence, whereas "no visible delimiters" settings recover from low initial rewards.

**(b) Length Scalability: Train short, generalize long.** Table data is inherently scalable in length. To investigate the effect of length scalability of table data on long-context reasoning, we conduct RL training with input lengths of 4k, 8k, and 16k. The results show consistent performance improvements across multiple long-context benchmarks as the length of structured table data scales up. Specifically, for Deepseek-R1-Distill-Qwen-32B, we observe relative gains of 0.48% and 1.39% at 8k and 16k, respectively, compared with the 4k experiment in Figure 8. The 14B results are similar, with details in Appendix D.

Moreover, we investigate the impact of length scaling on accuracy across different benchmark length ranges in Appendix E.2. The results show that TableLong, when trained on sequences within 16k, generalizes effectively to contexts beyond 16k and even up to 128k, with performance further improving as the training length scales.

**(c) Significantly improved long-context retrieval.** We adopt the Needle-in-a-Haystack benchmark to evaluate the gains of our TableLong in long-context retrieval, which measures a model's ability to locate a "needle" embedded at different depths within long documents.

As shown in Figure 3, compared with Deepseek-R1-Distill-Qwen-14B, our TableLong significantly improves long-context retrieval performance by 31.60%, increasing the retrieval score from 69.30% to 91.20%. Moreover, compared with Deepseek-R1-Distill-Qwen-32B, the retrieval score increases by 13.02%, from 87.95% to 99.40%, approaching near-perfect retrieval performance. These results indicate that structured table data, potentially benefiting from its periodic non-vanishing structural properties, can substantially enhance long-context retrieval for RL-based post-training.

## 4.3. Decomposing Tabular Capabilities for Long-Context Reasoning

In this subsection, we further decompose the capabilities of tabular data from three perspectives, including structural properties, multi-hop reasoning, and grounding. We also systematically analyze their contributions to the long-context reasoning capabilities of LLMs, yielding several inspiring insights.

### 4.3.1. STRUCTURAL PROPERTY

To investigate the effect of the periodic non-vanishing structural properties inherent to table data, we design and conduct experiments from two perspectives: the semantics of table cell contents and the visible delimiter structures of tables, yielding the following insights:

> **Insight 1**
>
> The inherent structure of table data serves as a foundation for long-range dependencies, while semantic content provides the reasoning signal for complex long-context reasoning.

Specifically, we construct "no semantic" table data (Figure 11) and "no visible delimiters" table data (Figure 12), with details provided in Appendix F.

Figure 4(a) shows that, relative to the baseline, "no semantic" with simple instruction prompting achieves an average performance improvement of 1.67%, indicating that the enhancement in long-context reasoning mainly stems from the table's inherent organizational structure, namely its periodic non-vanishing structural properties.

Relative to "Ours", the "no semantic" with only simple instruction prompting converges rapidly (within approxi-

*Table 2.* **Performance comparison across varying scales of table cells.** Bold values indicate the best performance within each backbone.

| Model | LB-v2 | Loong | Brows | GSM-Inf | Oolong | Ruler | MRCR | Avg. |
|---|---|---|---|---|---|---|---|---|
| DS-R1-Distill-32B | | | | | | | | |
| \|− 0∼30 | 44.33 | 42.87 | 71.87 | **18.00** | 40.29 | 69.30 | 37.44 | 46.30 |
| \|− 0∼100 | 45.33 | 43.46 | 74.31 | 10.00 | 44.10 | **71.82** | 36.46 | 46.50 |
| \|− 0∼300+ | **45.73** | **45.30** | **74.31** | 14.80 | **51.41** | 66.41 | **40.57** | **48.36** |
| DS-R1-Distill-14B | | | | | | | | |
| \|− 0∼30 | 36.98 | 35.60 | 72.17 | 6.40 | 41.70 | **82.16** | 27.63 | 43.23 |
| \|− 0∼100 | 37.38 | **36.45** | 74.31 | 6.40 | 34.47 | 80.44 | 29.16 | 42.65 |
| \|− 0∼300+ | **38.97** | 35.44 | 74.31 | **9.60** | **45.95** | 80.72 | **30.48** | **45.07** |

*Table 3.* **Performance comparison across varying scales of table count.** Bold values indicate the best performance within each backbone.

| Model | LB-v2 | Loong | Brows | GSM-Inf | Oolong | Ruler | MRCR | Avg. |
|---|---|---|---|---|---|---|---|---|
| DS-R1-Distill-32B | | | | | | | | |
| \|− 1 | 44.93 | 43.65 | 73.39 | **16.80** | 50.78 | 62.97 | 34.10 | 46.66 |
| \|− 1∼5 | 45.33 | 44.27 | 73.37 | 15.00 | 49.63 | 64.37 | 37.79 | 47.11 |
| \|− 1∼30 | **45.73** | **45.30** | **74.31** | 14.80 | **51.41** | **66.41** | **40.57** | **48.36** |
| DS-R1-Distill-14B | | | | | | | | |
| \|− 1 | 34.39 | **37.05** | 71.25 | 6.20 | 42.09 | 77.03 | **31.59** | 42.80 |
| \|− 1∼5 | **39.56** | 36.24 | 72.78 | 6.80 | 41.38 | 79.08 | 30.48 | 43.76 |
| \|− 1∼30 | 38.97 | 35.44 | **74.31** | **9.60** | **45.95** | **80.72** | 30.48 | **45.07** |

mately 40 steps) in Figure 4(b), leading to an average performance drop of 6.57%. This indicates that both the semantic content of the table and the complexity of the instructions are also crucial for long-context reasoning.

> **Insight 2**
>
> The visible delimiter structures do not affect the inherent structural properties of tables during RL training.

Figure 4 shows that the "no visible delimiters" setting does not exhibit a significant performance drop compared to "Ours", with a reduction of about 0.77%, and exhibits a similar asymptotic performance. in the RL reward curve. This indicates that visible delimiter structures do not affect the intrinsic periodic non-vanishing structural properties of tables, which are the key factor.

Moreover, the lower initial reward and consistently inferior performance of "no visible delimiters" suggest that removing visible delimiters makes table data harder for LLMs to interpret during training.

We further randomly replace delimiters with noise text (illustrated in Figure 13) and observe results that are largely consistent with the "no visible delimiters" setting. This further demonstrates the robustness of table data for RL training.

### 4.3.2. MULTI-HOP REASONING

> **Insight 3**
>
> Table linearization induces multi-hop reasoning that enhances long-context reasoning, and increasingly multi-hop patterns further strengthen this effect.

We characterize the "multi-hop" reasoning behavior of LLMs by the number of table cells involved in instruction operations, particularly along the column dimension. Specifically, when a two-dimensional table is linearized into a one-dimensional token sequence $\phi(\mathcal{T})$, cells within the same column become widely separated, forcing LLMs to attend to non-adjacent tokens and perform "multi-hop" reasoning. Representative examples are shown in Figure 9, and further details are in Appendix F.

As shown in Table 2, the model maintains strong long-context reasoning performance even with a small number of involved table cells (46.30% on average for the 0∼30 range). Moreover, scaling up the number of involved cells further improves performance on OOD long-context reasoning benchmarks. These findings suggest that the "multi-hop" property induced by table linearization effectively enhances long-context reasoning, and that increasing task complexity and multi-hop behavior can further strengthen this capability.

### 4.3.3. GROUNDING

> **Insight 4**
>
> Tables in $\phi(\mathcal{T})$ provide "grounding", and scaling up quantity strengthens these signals, guiding attention to key tokens and improving long-context reasoning.

Similarly, the number of tables in the prompt reflects the "grounding" capability of LLMs. Specifically, multi-table instruction operations compel the model to distinguish tables distributed across long contexts and to **associate** natural language queries with specific tables, making these tables act as sources of "grounding".

The results show that even a single table enables strong long-context reasoning in Table 3. As the number of tables scales up, the model achieves better performance on OOD long-context reasoning benchmarks, with gains of +0.45% for 1∼5 tables and +1.7% for "all" tables compared to the single-table setting. This indicates that linearized tables in the token sequence $\phi(\mathcal{T})$ act as "grounding" elements that the model need to attend to. As their number scales up, the distributed grounding signals across $\phi(\mathcal{T})$ increase, forcing the model to learn to allocate attention to key tokens in the sequence, thereby further enhancing long-context reasoning capability.

*Table 4.* **Generalization on OOD Benchmarks.** We evaluate models on GPQA-Diamond (**GPQA**), AIME 2025 (**AIME**), MultiChallenge (**MC**), and LiveCodeBench (**LCB**). Shaded rows indicate our RL-finetuned models (**+ Ours**). Our method achieves significant gains, notably **+11.9%** on LCB for the 32B model.

| Model | GPQA | AIME | MC | LCB |
|---|---|---|---|---|
| DS-R1-Distill-32B | 56.06 | 60.00 | 30.28 | 46.71 |
| DS-R1-Distill-32B **+ Ours** | **63.64** | **70.00** | **32.97** | **58.68** |
| DS-R1-Distill-14B | 55.56 | 43.33 | 23.08 | 43.71 |
| DS-R1-Distill-14B **+ Ours** | **59.60** | **46.67** | **26.86** | **45.51** |

### 4.4. Generalization Results to Other Domains

In this subsection, we evaluate the performance on out-of-domain (OOD) benchmarks from diverse fields, as shown in Table 4. Specifically, we consider GPQA-Diamond (science), AIME 2025 (math), MultiChallenge (multi-turn dialogue), and LiveCodeBench (code) to assess the generalization ability of TableLong across heterogeneous OOD domains. The results demonstrate that for Deepseek-R1-Distill-Qwen-32B, models trained with TableLong achieve substantial improvements of +7.58%, +10.00%, +2.69%, and +11.97% on these benchmarks, respectively. These significant gains indicate that TableLong strengthens their long-context retrieval and reasoning abilities, enabling effective generalization to a wide range of OOD domains.

## 5. Related Work

**Long-context Reasoning.** As real-world tasks become increasingly complex, long-context capability has become a crucial competency for large language models (LLMs). Early efforts on context length extension mainly focused on pre-training and mid-training (Xiong et al., 2024; Gao et al., 2025; Wang et al., 2025b), which improved long-context retrieval but did not effectively enable long-context reasoning. With the emergence of Deepseek R1 (DeepSeek-AI, 2025), reinforcement learning (RL) has become a key paradigm for enhancing LLM reasoning ability. Specifically, QwenLong L1 (Wan et al., 2025; Shen et al., 2025) pioneered long-context reasoning RL, catalyzing subsequent progress in both methodologies (Yan et al., 2025; Yang et al., 2025c; Zhu et al., 2025a; Lin et al., 2025) and benchmarks (Bai et al., 2025; Ling et al., 2025; Li et al., 2024a; Chen et al., 2026; Li et al., 2024c; Wang et al., 2025a). These developments have significantly accelerated the enhancement of long-context capabilities in LLMs.

**Data Synthesis Pipeline.** Moreover, long-context reasoning RL requires high-quality data, motivating increasing interest in data synthesis pipelines (Zhu et al., 2025b; Yang et al., 2025b; Zhang et al., 2025). The QwenLong series further improves dataset scale, diversity, and complexity through iterative pipeline design (Wan et al., 2025; Shen et al., 2025). However, few studies systematically investigate what types of data are effective for long-context reasoning and why, which is crucial for the targeted construction of long-context reasoning RL datasets. In this work, we study the effectiveness of table data for long-context reasoning and analyze the underlying mechanisms, and further provide a complete table data synthesis pipeline for the community. In parallel, we also examine the impact of document-style tasks on long-context reasoning in another work.

## 6. Conclusion and Future Work

In this work, we systematically investigate why scalable table data enhances long-context reasoning. Based on these, we propose a simple and scalable pipeline, namely Table-Long, for synthesizing high-quality, diverse, and verifiable structured table data to improve long-context reasoning via RL. Specifically, we mathematically analyze tabular dependency structures using mutual information, revealing periodic non-vanishing dependencies in table data, and experimentally validate the underlying mechanisms with comprehensive empirical studies. Extensive results show that table data significantly improves long-context reasoning across multiple benchmarks and generalizes well to diverse OOD tasks. In future work, we will further explore other data types to deepen data-centric understanding of long-context reasoning. We hope our findings provide practical guidance for effective post-training data design in LLMs.

## Impact Statement

This paper presents work whose goal is to advance the field of Machine Learning. There are many potential societal consequences of our work, none which we feel must be specifically highlighted here.

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

# Appendix

# A. Theoretical Details and Proofs

### A.1. Information-Theoretic Preliminaries

**Definition A.1** (Mutual Information (Cover, 1999)). For random variables $X$ and $Y$ with joint distribution $P_{X,Y}$ and marginals $P_X$, $P_Y$, the mutual information is:

$$I(X;Y) = D_{KL}(P_{X,Y} \| P_X \otimes P_Y) = \sum_{x,y} P(x,y) \log \frac{P(x,y)}{P(x)P(y)} \tag{7}$$

Mutual information is symmetric, non-negative, and equals zero if and only if $X \perp Y$.

**Definition A.2** (Kullback-Leibler Divergence). For distributions $P$ and $Q$ over the same space:

$$D_{KL}(P \| Q) = \sum_x P(x) \log \frac{P(x)}{Q(x)} \tag{8}$$

$D_{KL}(P \| Q) \geq 0$ with equality if and only if $P = Q$.

**Definition A.3** (Mixture Distribution). Given column distributions $\{P_j\}_{j=1}^m$, the mixture distribution is:

$$Q(v) = \frac{1}{m} \sum_{j=1}^m P_j(v) \tag{9}$$

This is the marginal distribution of a cell $T_{i,J}$ when $J \sim \text{Uniform}([m])$.

**Definition A.4** (Position-to-Column Mapping). For positions in the data region of a linearized table ($t > m$), define:

$$\text{col}(t) = ((t - m - 1) \mod m) + 1 \in [m] \tag{10}$$

### A.2. Assumptions

**Assumption A.5** (Column Semantic Consistency). All cells in column $j$ are drawn from the same distribution:

$$T_{i,j} \sim P_j, \quad \forall i \in [n], \forall j \in [m] \tag{11}$$

**Assumption A.6** (Column Distribution Distinctiveness). Different columns have statistically distinguishable distributions:

$$D_{KL}(P_j \| P_k) > 0, \quad \forall j \neq k \tag{12}$$

Equivalently, for each pair $j \neq k$, there exists some value $v$ such that $P_j(v) \neq P_k(v)$.

*Remark* A.7 (On Intra-Column Independence). We do **not** assume $T_{i,j} \perp T_{k,j} \mid j$ (conditional independence within columns). Real tables often exhibit intra-column dependencies (e.g., sorted data, time series). Our proofs analyze the conditional independence case to establish a *lower bound* on $I^{same}$; any intra-column dependency strictly increases the mutual information.

### A.3. Some Lemmas

**Lemma A.8** (Same-Column Pair Mutual Information). *Let $J \sim Uniform([m])$. Under Assumptions A.5 and A.6:*

*(i)* $I^{same} := I(T_{1,J}; T_{2,J}) > 0$

*(ii) For any $i \neq k$: $I(T_{i,J}; T_{k,J}) = I^{same}$*

*That is, the same-column mutual information is strictly positive and independent of row indices.*

*Proof.* We prove both parts simultaneously.

**Step 1: Marginal distributions.** For any row index $i$, by Assumption A.5 and $J \sim \text{Uniform}([m])$:

$$P(T_{i,J} = a) = \sum_{j=1}^{m} P(J = j)P_j(a) = \frac{1}{m} \sum_{j=1}^{m} P_j(a) = Q(a) \tag{13}$$

This holds for all $i$, so all $T_{i,J}$ share the same marginal distribution $Q$.

**Step 2: Joint distribution.** For any $i \neq k$, conditioning on the column:

$$P(T_{i,J} = a, T_{k,J} = b) = \sum_{j=1}^{m} P(J = j) \cdot P(T_{i,j} = a, T_{k,j} = b \mid J = j) \tag{14}$$

To establish a lower bound independent of intra-column structure, consider the case where $T_{i,j} \perp T_{k,j} \mid j$. Then:

$$P(T_{i,J} = a, T_{k,J} = b) = \frac{1}{m} \sum_{j=1}^{m} P_j(a)P_j(b) =: R(a,b) \tag{15}$$

Note that $R(a,b)$ depends only on the column distributions $\{P_j\}$, not on $i$ or $k$. If $T_{i,j} \not\perp T_{k,j} \mid j$, the actual joint distribution differs from $R$, but we will show $R \neq Q \otimes Q$, which implies $I(T_{i,J}; T_{k,J}) > 0$ in all cases.

**Step 3: Comparison with product of marginals.** Under independence, we would have:

$$Q(a)Q(b) = \frac{1}{m^2} \sum_{j=1}^{m} \sum_{l=1}^{m} P_j(a)P_l(b) \tag{16}$$

Consider the diagonal $a = b$:

$$
\begin{aligned}
R(a,a) - Q(a)^2 &= \frac{1}{m} \sum_{j=1}^{m} P_j(a)^2 - \frac{1}{m^2} \left( \sum_{j=1}^{m} P_j(a) \right)^2 \\
&= \frac{1}{m^2} \left[ m \sum_{j=1}^{m} P_j(a)^2 - \left( \sum_{j=1}^{m} P_j(a) \right)^2 \right] \\
&= \frac{\text{Var}_j[P_j(a)]}{m}
\end{aligned}
\tag{17}
$$

where $\text{Var}_j[P_j(a)] = \frac{1}{m} \sum_j P_j(a)^2 - \left( \frac{1}{m} \sum_j P_j(a) \right)^2$ is the variance of $P_j(a)$ across columns.

**Step 4: Positivity.** By Assumption A.6, there exist $j \neq l$ with $P_j \neq P_l$. Hence there exists some $a$ with $P_j(a) \neq P_l(a)$, implying $\text{Var}_j[P_j(a)] > 0$.

Therefore $R(a,a) > Q(a)^2$ for some $a$, so $R \neq Q \otimes Q$.

**Step 5: Conclusion.** Since the joint distribution differs from $Q \otimes Q$:

$$I(T_{i,J}; T_{k,J}) = D_{KL}(P_{T_{i,J}, T_{k,J}} \| Q \otimes Q) > 0 \tag{18}$$

Moreover, under conditional independence, the joint is exactly $R(a,b)$ from (15), which is independent of $i, k$. Thus $I(T_{i,J}; T_{k,J})$ is constant for all $i \neq k$, proving part (ii). $\qquad\square$

**Lemma A.9** (Periodic Structure)**.** *For a table with $n$ rows and $m$ columns, and any $k \in \{1, \ldots, n-1\}$:*

$$\bar{I}_{table}(km) = I^{same} \tag{19}$$

*Proof.* **Step 1: Same-column positions.** For $t, t' > m$ in the data region:

$$\text{col}(t) = \text{col}(t') \iff t' - t \equiv 0 \pmod{m} \tag{20}$$

**Step 2: Structure at lag $km$.** When $d = km$ with $1 \le k \le n - 1$, for each $t \in \mathcal{S}_{km} = \{t : m < t \le L - km\}$:

- $W_t = T_{r_1, c}$ for some row $r_1$ and column $c = \text{col}(t)$

- $W_{t+km} = T_{r_1 + k, c}$ (same column, row differs by $k$)

By Lemma A.8(ii):

$$I(W_t; W_{t+km}) = I(T_{r_1, c}; T_{r_1 + k, c}) = I^{same} \tag{21}$$

**Step 3: Averaging.** Since every pair contributes equally:

$$\bar{I}_{table}(km) = \frac{1}{|\mathcal{S}_{km}|} \sum_{t \in \mathcal{S}_{km}} I^{same} = I^{same} \tag{22}$$

$\square$

**Lemma A.10** (Cross-Column Independence). *Let $J, L \sim \text{Uniform}([m])$ be independent. Then $T_{i,J} \perp T_{k,L}$, and hence $I(T_{i,J}; T_{k,L}) = 0$.*

*Proof.* When $J \perp L$:

$$P(T_{i,J} = a, T_{k,L} = b) = \sum_{j,l} P(J = j) P(L = l) P_j(a) P_l(b)$$

$$= \left( \frac{1}{m} \sum_j P_j(a) \right) \left( \frac{1}{m} \sum_l P_l(b) \right) = Q(a) Q(b) \tag{23}$$

Hence $T_{i,J} \perp T_{k,L}$. $\square$

## A.4. Characterization of $I^{same}$

**Proposition A.11** (Characterization of Same-Column MI). *Under Assumptions A.5 and A.6, in the conditional independence case ($T_{i,j} \perp T_{k,j} \mid j$):*

$$I^{same} = D_{KL}(R \| Q \otimes Q) \tag{24}$$

*where $R(a, b) = \frac{1}{m} \sum_{j=1}^m P_j(a) P_j(b)$ and $Q = \frac{1}{m} \sum_j P_j$.*

*The magnitude of $I^{same}$ is governed by the **cross-column variance**:*

$$\sigma^2 := \sum_a Var_j[P_j(a)] = \sum_a \left( \frac{1}{m} \sum_j P_j(a)^2 - Q(a)^2 \right) \tag{25}$$

*Specifically:*

- *$I^{same} > 0$ if and only if $\sigma^2 > 0$ (i.e., columns are not all identical)*

- *$I^{same}$ increases monotonically with $\sigma^2$*

- *When columns are highly distinct (large $\sigma^2$), $I^{same}$ is correspondingly large*

*Proof.* The expression $I^{same} = D_{KL}(R\|Q \otimes Q)$ follows directly from the proof of Lemma A.8.

For the characterization, note from (17) that $R(a, a) - Q(a)^2 = \frac{\text{Var}_j[P_j(a)]}{m}$. The KL divergence $D_{KL}(R\|Q \otimes Q)$ is determined by how much $R$ differs from $Q \otimes Q$. Since the diagonal differences $R(a, a) - Q(a)^2$ are proportional to $\text{Var}_j[P_j(a)]$, the total cross-column variance $\sigma^2 = \sum_a \text{Var}_j[P_j(a)]$ governs the magnitude of $I^{same}$.

By Assumption A.6, $\sigma^2 > 0$, ensuring $I^{same} > 0$. $\square$

*Remark* A.12. The cross-column variance $\sigma^2$ can be interpreted as a measure of column heterogeneity. For example:

- If all columns have the same distribution ($P_j = P$ for all $j$), then $\sigma^2 = 0$ and Assumption A.6 is violated.

- If columns have disjoint supports (e.g., column 1 contains only integers, column 2 contains only strings), then $\sigma^2$ is maximal and $I^{same}$ is large.

## A.5. Proofs of Main Theorems

### A.5.1. PROOF OF THEOREM 2.2

*Proof.* This follows directly from Lemma A.9: for any $k \in \{1, \ldots, n-1\}$, $\bar{I}_{table}(km) = I^{same} > 0$. $\square$

### A.5.2. PROOF OF COROLLARY 2.3

*Proof.* For any $K \in \mathbb{Z}^+$, consider a table with $n > K$ rows. By Theorem 2.2, $\bar{I}_{table}(Km) = I^{same}$. Since $K$ is arbitrary, the set $\{km : k \in \mathbb{Z}^+\}$ is unbounded, and at each such lag the average MI equals $I^{same}$. Therefore:

$$\liminf_{d \to \infty} \bar{I}_{table}(d) \geq I^{same} > 0 \tag{26}$$

$\square$

### A.5.3. PROOF OF COROLLARY 2.4

*Proof.* By Theorem 2.2, $\bar{I}_{table}(km) = I^{same} > 0$ for all $k \in \mathbb{Z}^+$. By the power-law assumption (1), $I_{text}(km) \sim C(km)^{-\alpha} \to 0$ as $k \to \infty$. Hence:

$$\lim_{k \to \infty} \frac{\bar{I}_{table}(km)}{I_{text}(km)} = \lim_{k \to \infty} \frac{I^{same}}{C(km)^{-\alpha}} = +\infty \tag{27}$$

$\square$

### A.5.4. PROOF OF THEOREM 2.6

*Proof.* **Part 1 (Table data):** For any $\tau < I^{same}$ and any $K \in \mathbb{Z}^+$, consider a table with $n > K$ rows. By Lemma A.9:

$$\bar{I}_{table}(Km) = I^{same} > \tau \tag{28}$$

Since $K$ is arbitrary:

$$D_{eff}^{table}(\tau) = \sup\{d : \bar{I}_{table}(d) \geq \tau\} = +\infty \tag{29}$$

**Part 2 (Natural language):** Under power-law decay $I_{text}(d) \sim Cd^{-\alpha}$, for any $\tau > 0$:

$$I_{text}(d) < \tau \quad \text{for all } d > d^* := (C/\tau)^{1/\alpha} \tag{30}$$

Hence $D_{eff}^{text}(\tau) \leq d^* < \infty$. $\square$

## A.6. Proofs of Propositions

### A.6.1. Proof of Proposition for Uniform Attention Requirements

*Proof.* Let $W_t$ be a cell in column $c = \text{col}(t)$. The set of past tokens in the same column is $\mathcal{C}_t := \{W_s : s < t, \text{col}(s) = c\}$.

For any $W_s \in \mathcal{C}_t$, both $W_s$ and $W_t$ are cells from column $c$, at different rows. By Lemma A.8(ii):

$$I(W_s; W_t) = I^{same} > 0 \tag{31}$$

regardless of the distance $t - s$. Thus all same-column predecessors provide the same amount of information $I^{same}$ about $W_t$, requiring uniform attention across the context. $\square$

### A.6.2. Proof of Proposition for Context Length Requirements

*Proof.* Consider any column $c \in [m]$. The cell at row 1 occupies position $m + c$ in the linearized sequence, while the cell at row $n$ occupies position $m + (n - 1)m + c = m + c + (n - 1)m$. Thus these two same-column cells are separated by exactly $(n - 1) \cdot m$ positions.

By Lemma A.8, this pair has mutual information $I^{same} > 0$. To capture this dependency, the model's effective context must span at least $(n - 1) \cdot m$ tokens. $\square$

## A.7. Auxiliary Result

**Lemma A.13** (Positive Column-Type MI). *Let $J \sim Uniform([m])$ and $T_J$ be a cell from column $J$. Under Assumptions A.5– A.6:*

$$I(T_J; J) > 0 \tag{32}$$

*Proof.* We have $I(T_J; J) = H(J) - H(J|T_J) = \log m - H(J|T_J)$.

By Bayes' rule: $P(J = j | T_J = v) = \frac{m P_j(v)}{\sum_l P_l(v)}$.

By Assumption A.6, there exist $j \neq k$ and $v$ with $P_j(v) \neq P_k(v)$, so $P(J = j | T_J = v) \neq P(J = k | T_J = v)$. The posterior is non-uniform for some $v$, implying $H(J|T_J) < H(J)$, hence $I(T_J; J) > 0$. $\square$

# B. Training Details

All experiments in this paper are performed on 64 H20 NVIDIA GPUs with the Verl framework (Sheng et al., 2024) based on the GRPO algorithm (Shao et al., 2024), as detailed below:

$$\mathcal{J}_{\text{GRPO}}(\theta) = \mathbb{E}_{(q,a) \sim \mathcal{D}, \{o_i\}_{i=1}^G \sim \pi_{\theta_{\text{old}}}(\cdot|q)}$$
$$\left[ \frac{1}{G} \sum_{i=1}^G \frac{1}{|o_i|} \sum_{t=1}^{|o_i|} \left( \min \left( r_{i,t}(\theta) \hat{A}_{i,t}, \text{clip}(r_{i,t}(\theta), 1 - \epsilon, 1 + \epsilon) \hat{A}_{i,t} \right) - \beta D_{\text{KL}}(\pi_\theta \| \pi_{\text{ref}}) \right) \right] \tag{33}$$

where,

$$r_{i,t}(\theta) = \frac{\pi_\theta(o_{i,t} \mid q, o_{i,<t})}{\pi_{\theta_{\text{old}}}(o_{i,t} \mid q, o_{i,<t})}, \quad \hat{A}_{i,t} = \frac{R_i - \text{mean}(\{R_i\}_{i=1}^G)}{\text{std}(\{R_i\}_{i=1}^G)}. \tag{34}$$

- *Rollout Strategy:* We configure the model for long-context input processing. To ensure sample efficiency, we implement dynamic sampling, performing large-scale rollouts to identify and retain only informative groups with non-zero advantage variance (i.e., filtering out unanimously correct or incorrect samples).

- *Optimization Algorithm:* We adopt the clip higher strategy (Yu et al., 2025). This strategy is crucial for accelerating convergence on positive signals while ensuring update safety on negative ones.

- *Reward Modeling:* We utilize gpt-oss-120B(OpenAI, 2025c) as an outcome-based judge to verify the semantic consistency between generated responses and the ground truth.

Detailed hyperparameters (e.g., batch sizes, learning rates, and clipping thresholds) are provided in Table 5.

*Table 5.* Detailed hyperparameters for RL training

| Hyperparameter | Value |
|---|---|
| ***Computational Setup*** | |
| Hardware | $64 \times$ NVIDIA H20 |
| Parallelism Strategy | TP=8 |
| Offload | Disabled |
| ***Training Parameters*** | |
| Max Sequence Capacity | 32,768 |
| Max Input Length | 28,672 |
| Max Output Length | 4,096 |
| Global Train Batch Size | 128 |
| Mini-Batch Size | 32 |
| Learning Rate | $2 \times 10^{-6}$ |
| KL Coefficient ($\beta$) | 0.001 |
| Rollout Batch Size (Gen BS) | 256 |
| Group Size ($G$) | 16 |
| Temperature | 0.85 |
| Top-$p$ | 1.0 (Disabled) |
| Clip Ratio High ($\epsilon_{high}$) | 0.28 |
| Clip Ratio Low ($\epsilon_{low}$) | 0.20 |

## C. Benchmark Specifications

To ensure a rigorous and standardized evaluation, we strictly adhere to the official protocols of each benchmark. **Evaluation:** We utilize the official scoring scripts provided by the respective benchmarks. The detailed composition of our evaluation suite is provided in Table 6 and 7.

*Table 6.* Details of the evaluation benchmarks for long-context reasoning. "Subset/Task" indicates the specific evaluation partition used. "Num. Samples" denotes the number of test instances after filtering.

| Category | Benchmark | Subset / Task | Num. Samples | Domain Focus |
|---|---|---|---|---|
| **Holistic** | LongBench-v2 (Bai et al., 2025) | *Full Set* | 503 | General Real-world |
| | Loong (Wang et al., 2024) | *Full Set* | 1,441 | Spotlight QA / Retrieval |
| **Retrieval** | Browscomplong (OpenAI, 2025a) | *Standard* | 327 | Complex Search |
| | MRCR (OpenAI, 2025b) | *Standard* | 1,437 | Multi-round QA |
| | Ruler (Hsieh et al., 2024) | *QA2 Task* | 2,495 | Variable Tracking |
| **Reasoning** | GSM-Infinite (Zhou et al., 2025) | *Standard* | 500 | Long-range Math |
| | Oolong (Bertsch et al., 2025) | *Synthetic Subset* | 3,127 | Info Aggregation |

*Table 7.* Details of the evaluation benchmarks for other domains. "Num. Samples" denotes the number of test instances after filtering.

| Category | Benchmark | Num. Samples | Domain Focus |
|---|---|---|---|
| Science | GPQA-Diamond (Rein et al., 2023) | 198 | Scientific Reasoning |
| Math | AIME 2025 (Balunović et al., 2025) | 30 | Mathematical Reasoning |
| Multi-turn Dialogue | MultiChallenge (Deshpande et al., 2025) | 273 | Real-world Multi-turn Dialogue |
| Code | LiveCodeBench (Jain et al., 2024) | 167 | Code Generation |

## D. Detailed Results.

In Figure 5, we provide additional decomposition experiments on semantics, delimiters, and noise for Deepseek-R1-Distill-Qwen-14B in Table 8. Additionally, we provide a detailed performance comparison across varying length scales for Deepseek-R1-Distill-Qwen-32B and 14B.

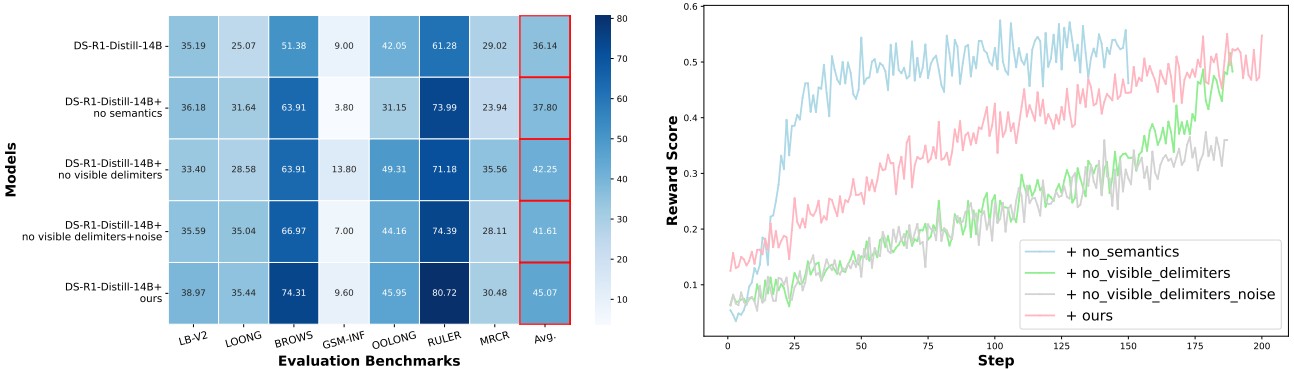

(a) Impact of semantics, delimiters, and noise.       (b) RL reward curves: Convergence and stability analysis.

*Figure 5.* **Decomposition experiments for DS-R1-Distill-14B.** (a) While structure alone ("no semantics") boosts baseline performance (+1.66%), semantics remain essential for peak results. Removing delimiters or adding **noise** yields negligible drops, confirming the primacy of intrinsic structure. (b) Models with "no semantics" suffer premature convergence, whereas "no visible delimiters" settings recover from low initial rewards.

*Table 8.* **Performance comparison across varying scales of length.** Bold values indicate the best performance within each backbone.

| Model | LB-V2 | LOONG | BROWS | GSM-INF | OOLONG | RULER | MRCR | Avg. |
|---|---|---|---|---|---|---|---|---|
| DS-R1-Distill-32B | | | | | | | | |
|   — 4k | 45.13 | 43.41 | **74.92** | 11.60 | 47.65 | **68.42** | 37.65 | 46.97 |
|   — 8k | **48.11** | 43.89 | 74.01 | 13.20 | 47.87 | 66.05 | 39.04 | 47.45 |
|   — 16k | 45.73 | **45.30** | 74.31 | **14.80** | **51.41** | 66.41 | **40.57** | **48.36** |
| DS-R1-Distill-14B | | | | | | | | |
|   — 4k | 36.78 | **36.03** | 74.31 | 8.00 | 43.81 | 80.08 | 30.27 | 44.18 |
|   — 8k | 35.79 | 35.93 | 72.78 | 6.40 | 43.52 | **81.80** | 29.30 | 43.64 |
|   — 16k | **38.97** | 35.44 | **74.31** | **9.60** | **45.95** | 80.72 | **30.48** | **45.06** |

## E. Detailed Results across Different Length Ranges on LongBench v2, MRCR and Oolong-Synth.

### E.1. Main Experiments

To further analyze the gains brought by our TableLong for long-context reasoning, we conduct an in-depth analysis of the main experimental results. Specifically, we consider three representative long-context benchmarks with different characteristics: LongBench-v2 (general real-world tasks), Oolong-Synth (information aggregation), and MRCR (multi-round QA), and evaluate performance across different length ranges. Notably, all our models are trained on table data within a 16k context window.

First, we clearly observe that for most benchmarks, model performance exhibits a gradual degradation as the evaluated length range increases. As shown in Figure 6(a), on the real-world LongBench-v2 benchmark, our TableLong achieves consistent and significant improvements (+3.92%) in the 16k+ length ranges across different backbones, demonstrating strong length extrapolation ability. This further indicates that table data can more effectively elicit long-context reasoning capability in larger models.

Moreover, as shown in Tables 9 and 10, on retrieval-oriented long-context benchmarks including Oolong-Synth and MRCR, models trained with our TableLong exhibit strong long-context retrieval capability. In particular, for the 64–128k length

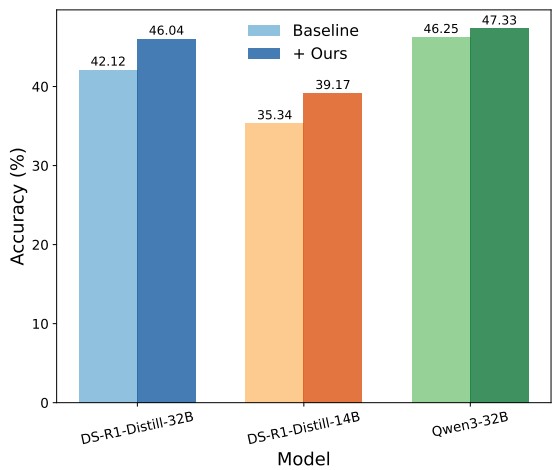

*(a)* Different backbone models VS. Ours.

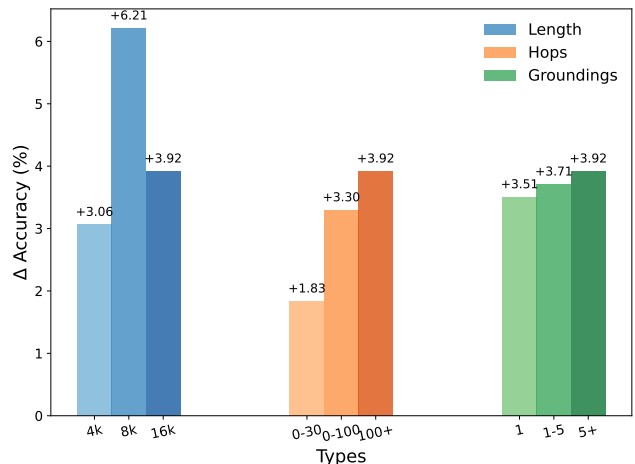

*(b)* Varying lengths, hops, and grounding.

*Figure 6.* **Performance comparison on the 16k+ length range of LongBench-v2.**

range, Deepseek-R1-Distill-Qwen-32B achieves notable improvements of +12.75% and +4.79%, respectively.

*Table 9.* Oolong-Synth performance across different length ranges

| Model | 0-4k | 4k-8k | 8k-16k | 16k-32k | 32k-64k | 64-128k |
|---|---|---|---|---|---|---|
| DS-R1-Distill-32B | 63.82 | 51.50 | 38.50 | 28.75 | 23.00 | 26.25 |
| DS-R1-Distill-32B + Ours | **69.49** | **62.75** | **55.50** | **45.25** | **38.50** | **39.00** |
| DS-R1-Distill-14B | 62.31 | 51.00 | 45.50 | 37.25 | 31.25 | 27.00 |
| DS-R1-Distill-14B + Ours | **66.30** | **52.50** | **47.75** | **40.00** | **33.50** | **34.75** |
| Qwen3-32B | 60.66 | 40.25 | 33.25 | 20.50 | 11.75 | 15.00 |
| Qwen3-32B + Ours | **63.69** | **53.75** | **53.00** | **35.75** | **28.25** | **24.25** |

*Table 10.* MRCR performance across different length ranges

| Model | 0-4k | 4k-8k | 8k-16k | 16k-32k | 32k-64k | 64k-128k |
|---|---|---|---|---|---|---|
| DS-R1-Distill-32B | - | 39.43 | 38.52 | 30.41 | 27.46 | 25.62 |
| DS-R1-Distill-32B + Ours | - | **50.81** | **47.04** | **40.54** | **36.61** | **30.31** |
| DS-R1-Distill-14B | - | 37.80 | 38.15 | 32.79 | 26.51 | 14.37 |
| DS-R1-Distill-14B + Ours | - | **44.31** | 34.07 | 28.04 | 25.76 | **22.81** |
| Qwen3-32B | - | 52.44 | 48.15 | 40.88 | 36.27 | 35.94 |
| Qwen3-32B + Ours | - | **52.44** | 45.93 | **42.23** | **41.36** | 33.44 |

### E.2. Scalability Experiments of Tabular Capabilities

**Length Scalability.** Similarly, as shown in Figure 6(b), on the real-world LongBench-v2 benchmark, the trained models with our TableLong exhibit progressive performance gains in the 16k+ length ranges as the input length of table data increases (+3.06%, +6.21%, and +3.92%, respectively). Consistent trends are also observed on Oolong-Synth and MRCR, as reported in Tables 11 and 12.

**Multi-hop Reasoning Scalability.** Specifically, we also observe that as the number of instruction-designed cells increases and the instructions become more complex, models trained with our TableLong exhibit progressively larger advantages on real-world long-context understanding and long-context retrieval tasks across different length ranges. As shown in

Figure 6(b), on the real-world LongBench-v2 benchmark, the trained models with our TableLong exhibit progressive performance gains in the 16k+ length ranges as the input length of table data increases (+1.83%, +3.30%, and +3.92%, respectively). Consistent trends are also observed on Oolong-Synth and MRCR, as reported in Tables 11 and 12.

**Grounding Scalability.** Similarly, as shown in Figure 6(b), on the real-world LongBench-v2 benchmark, the trained models with our TableLong exhibit progressive performance gains in the 16k+ length ranges as the input length of table data increases (+3.51%, +3.73%, and +3.92%, respectively). Consistent trends are also observed on Oolong-Synth and MRCR, as reported in Tables 11 and 12.

*Table 11.* Oolong performance comparison across context lengths.

| Model | 0-4k | 4k-8k | 8k-16k | 16k-32k | 32k-64k | 64-128k |
|---|---|---|---|---|---|---|
| DS-R1-Distill-32B | 63.82 | 51.50 | 38.50 | 28.75 | 23.00 | 26.25 |
| \|−4k | 70.70 | 61.50 | 52.75 | 39.50 | 31.25 | 31.75 |
| \|−8k | 70.01 | 61.50 | 57.25 | 45.75 | 32.00 | 28.25 |
| \|−0-30 | 68.64 | 56.75 | 34.75 | 29.75 | 21.50 | 23.00 |
| \|−0-100 | 72.94 | 60.75 | 47.50 | 32.00 | 25.50 | 25.50 |
| \|−1 | 71.39 | 64.75 | 56.50 | 42.75 | 34.00 | 33.50 |
| \|−1-5 | 71.11 | 64.00 | 57.50 | 47.75 | 35.00 | 32.25 |
| \|−Ours | 69.49 | 62.75 | 55.50 | 45.25 | **38.50** | **39.00** |

*Table 12.* MRCR performance comparison across context lengths with grouping.

| Group | Model | 0-4k | 4k-8k | 8k-16k | 16k-32k | 32k-64k | 64-128k |
|---|---|---|---|---|---|---|---|
| | DS-R1-Distill-32B | - | 39.43 | 38.52 | 30.41 | 27.46 | 25.62 |
| **Length** | \|−4k | - | 48.37 | 44.81 | 33.45 | 34.92 | 29.69 |
| | \|−8k | - | 51.63 | 43.33 | 37.84 | 34.24 | 31.87 |
| **Cells** | \|−0-30 | - | 48.78 | 42.95 | 36.15 | 33.90 | 27.50 |
| | \|−0-100 | - | 47.97 | 42.95 | 36.94 | 29.15 | 28.44 |
| **Tables** | \|−1 | - | 43.50 | 39.26 | 34.80 | 36.61 | 24.06 |
| | \|−1-5 | - | 49.19 | 40.37 | 35.47 | 36.61 | 29.38 |
| | \|−Ours | - | 50.81 | **47.04** | **40.54** | **36.61** | 30.31 |

# F. Details of Data Cases

---

## Prompt for SQL Generation

System Pormpt: You are a data generation expert proficient in SQL. Your task is to generate a standard SQL query and its corresponding Natural Language Question (NLQ) based on the provided schema and instructions.

Core Rules:

1. Table Referencing: Always use real table names from the schema; no placeholders.

2. NLQ Requirements:
   - Contextual: Questions must reflect the business context.
   - Natural Tone: Use colloquial or business language. No SQL jargon (e.g., "calculate average"). Test intent understanding, not translation.
   - Column Mapping: Explicitly map obscure column names (e.g., `col_x`) to their meanings in the question.

3. Output Format: Return ONLY a JSON object: `{"sql": "...", "natural_language_question": "..."}`.

Input Data: [Table List], [Schema], [Example Data], [Difficulty Instruction]

*Figure 7.* The instruction prompt used for generating SQL-NLQ pairs, which constrains SQL logic while eliciting natural, task-oriented questions.

---

**Question Type: Precise Retrieval**

**Question:**
Can you tell me the details of the game where the team played against the Houston Oilers?

**Answer:**

```
| Week | Date | Opponent | Location | Time ( ET ) | Result | Record |
| --- | --- | --- | --- | --- | --- | --- |
| 13 | Sun.  Dec.  3 | Houston Oilers | Three Rivers Stadium | 1:00pm | L 23{16 | 6{7 |
```

---

**Table:**

```
Table_1
| Week | Date | Opponent | Location | Time ( ET ) | Result | Record |
| --- | --- | --- | --- | --- | --- | --- |
| 1 | Sun.  Sep.  10 | Cleveland Browns | Three Rivers Stadium | 4:00pm | L 51{0 | 0{1 |
| 2 | Sun.  Sep.  17 | at Cincinnati Bengals | Riverfront Stadium | 1:00pm | L 41{10 | 0{2 |
| 3 | Sun.  Sep.  24 | Minnesota Vikings | Three Rivers Stadium | 1:00pm | W 27{14 | 1{2 |
| 4 | Sun.  Oct.  1 | at Detroit Lions | Pontiac Silverdome | 1:00pm | W 23{3 | 2{2 |
| 5 | Sun.  Oct.  8 | Cincinnati Bengals | Three Rivers Stadium | 1:00pm | L 26{16 | 2{3 |
| 6 | Sun.  Oct.  15 | at Cleveland Browns | Cleveland Municipal Stadium | 4:00pm | W 17{7 | 3{3 |
| 7 | Sun.  Oct.  22 | at Houston Oilers | Astrodome | 1:00pm | L 27{0 | 3{4 |
| 8 | Sun.  Oct.  29 | Kansas City Chiefs | Three Rivers Stadium | 1:00pm | W 23{17 | 4{4 |
| 9 | Sun.  Nov.  5 | at Denver Broncos | Mile High Stadium | 4:00pm | L 34{7 | 4{5 |
| 10 | Sun.  Nov.  12 | Chicago Bears | Three Rivers Stadium | 1:00pm9 | San Diego Chargers | Three Rivers
Stadium | 1:00pm | W 20{17 | 5{6 |
| 12 | Sun.  Nov.  26 | at Miami Dolphins | Joe Robbie Stadium | 1:00pm | W 34{14 | 6{6 |
| 13 | Sun.  Dec.  3 | Houston Oilers | Three Rivers Stadium | 1:00pm | L 23{16 | 6{7 |
| 14 | Sun.  Dec.  10 | at New York Jets | Giants Stadium | 1:00pm | W 13{0 | 7{7 |
| 15 | Sun.  Dec.  17 | New England Patriots | Three Rivers Stadium | 1:00pm | W 28{10 | 8{7 |
| 16 | Sun.  Dec.  24 | at Tampa Bay Buccaneers | Tampa Stadium | 1:00pm | W 31-22 | 9{7 |
```

*Figure 8.* An instance of the precise retrieval grounding task.

---

**Question Type: Multi-hop Reasoning**

**Question:**
Which stadiums had an average of over 13,000 fans attending each match played there?

**Answer:**

```
| Venue | avg_attendance |
| --- | --- |
| Malmö Stadion | 15976.0 |
| Olympia | 13385.0 |
| Råsunda | 25983.2 |
```

---

**Table:**

```
Table
| Date | Venue | Opponents | Score | Comp | Attendance |
| --- | --- | --- | --- | --- | --- |
| 2007-04-07 | Råsunda | BP | 0{1 | Allsv. | 15 092 |
| 2007-04-16 | Stadion | Halmstad | 2{0 | Allsv. | 10 747 |
| 2007-04-21 | Stadion | Helsingborg | 3{1 | Allsv. | 11 223 |
| 2007-04-29 | Fredriksskans | Kalmar | 0{1 | Allsv. | 7 349 |
| 2007-05-07 | Borås Arena | Elfsborg | 2{2 | Allsv. | 11 533 |
| 2007-05-14 | Stadion | GAIS | 0{1 | Allsv. | 10 122 |
| 2007-05-21 | Vångavallen | Trelleborg | 3{0 | Allsv. | 3 012 |
| 2007-05-28 | Råsunda | AIK | 3{1 | Allsv. | 32 529 |
| 2007-06-12 | Stadion | Göteborg | 2{1 | Allsv. | 12 697 |
| 2007-06-19 | Råsunda | Hammarby | 0{2 | Allsv. | 23 545 |
| 2007-06-25 | Behrn Arena | Örebro | 0{0 | Allsv. | 11 565 |
| 2007-07-02 | Stadion | Gefle | 2{1 | Allsv. | 12 030 |
| 2007-07-07 | Stadion | Malmö | 1{0 | Allsv. | 11 515 |
| 2007-07-17 | Malmö Stadion | Malmö | 1{1 | Allsv. | 15 976 |
| 2007-07-21 | Stadion | Elfsborg | 2{1 | Allsv GAIS | 1{1 | Allsv. | 6 453 |
| 2007-08-13 | Råsunda | Hammarby | 1{0 | Allsv. | 24 634 |
| 2007-08-16 | Nya Ullevi | Göteborg | 1{1 | Allsv. | 12 187 |
| 2007-08-27 | Stadion | Kalmar | 1{3 | Allsv. | 13 387 |
| 2007-09-03 | Olympia | Helsingborg | 4{1 | Allsv. | 13 385 |
| 2007-09-17 | Stadion | Trelleborg | 1{1 | Allsv. | 9 700 |
| 2007-09-24 | Råsunda | AIK | 1{1 | Allsv. | 34 116 |
| 2007-09-29 | Stadion | Örebro | 4{1 | Allsv. | 9 311 |
| 2007-10-07 | Strömvallen | Gefle | 2{0 | Allsv. | 7 130 |
| 2007-10-22 | Örjans Vall | Halmstad | 2{1 | Allsv. | 7 147 |
| 2007-10-28 | Stadion | BP | 0{1 | Allsv. | 14 222 |
```

*Figure 9.* An instance of the multi-hop reasoning task.

**Question Type: Multi-table Grounding**

**Question:**

For each constituency in the table_3_election results where the winner's party has an entry in the party performance summary table, list the constituency name, the winner's name and party, how many total seats that winning party won nationwide, the runner-up's name and party, and how many total seats the runner-up's party won nationwide

**Answer:**

```
| Constituency | Winner_Name | Winner_Party | Winner_Party_Total_Seats_Won | Runner_Up_Name | Runner_Up_Party
| Runner_Up_Party_Total_Seats_Won |
| --- | --- | --- | --- | --- | --- | --- |
| 1.  Chennai North | D. Pandian | Indian National Congress | 10 | Aladi Aruna | DMK | |
| 2.  Chennai Central | Era Anbarasu c | Indian National Congress | 10 | N. V. N. Somu | DMK | |
| 4.  Sriperumbudur | Margatham Chandrasekar c | Indian National Congress | 10 | K. Sundaram | DMK | |
| 6.  Arakkonam | R. Jeevarathinam c | Indian National Congress | 10 | M. Kannaiyan | DMK | |
| 7.  Vellore | B. Akber Pasha | Indian National Congress | 10 | P. Shanmugam | DMK | |
```

---

**Table:**

```
Table_1
| Constituency | Winner | Party | Margin | Runner-up a | Party a |
| --- | --- | --- | --- | --- | --- |
| 1.  Chennai North | D. Pandian | Indian National Congress | 118518 | Aladi Aruna | DMK |
| 2.  Chennai Central | Era Anbarasu c | Indian National Congress | 103271 | N. V. N. Somu | DMK |
| 3.  Chennai South | R. Sridharan | Anna Dravida Munnetra Kazhagam | 162528 | T. R. Balu | DMK |
⋮
⋮

Table_2
| Constituency | Winner | Party | Margin | Runner-up a | Party a |
| --- | --- | --- | --- | --- | --- |
| 1.  Chennai North | D. Pandian | Indian National Congress | 118518 | Aladi Aruna | DMK |
| 2.  Chennai Central | Era Anbarasu c | Indian National Congress | 103271 | N. V. N. Somu | DMK |
| 3.  Chennai South | R. Sridharan | Anna Dravida Munnetra Kazhagam | 162528 | T. R. Balu | DMK |
⋮
⋮

Table_3
| Constituency | Winner | Party | Margin | Runner-up a | Party a |
| --- | --- | --- | --- | --- | --- |
| 1.  Chennai North | D. Pandian | Indian National Congress | 118518 | Aladi Aruna | DMK |
| 2.  Chennai Central | Era Anbarasu c | Indian National Congress | 103271 | N. V. N. Somu | DMK |
| 3.  Chennai South | R. Sridharan | Anna Dravida Munnetra Kazhagam | 162528 | T. R. Balu | DMK |
⋮
⋮

[N complete tables abbreviated]
```

*Figure 10.* An instance of the multi-table grounding task.

---

**Question Type: No Semantic**

**Question:**
What is the element in row "born" and column "people" in Table 2?
Note: The first row contains column names, and the first column contains row names.
Only output the answer, do not output any other irrelevant content.

**Answer:**

```
walking
```

---

**Table:**

```
Table 1 - County:
| | Tubby | for | account | behalf | four | igneous | 35th | The |
| --- | --- | --- | --- | --- | --- | --- | --- | --- |
| number | Haim | meant | the | was | many | was | grabbed | when |
| top | Nine | September | the | work | local | that | Players | recorded |
| they | that | cites | paraboloid | Caribbean | SSAU | hardhitting | town | protein |
| which | Though | produced | Kepler | The | school | supported | the | 1972 |
| Population | University | complementary | dramas | the | tracked | drove | approved | impedance |

Table 2 - people:
| | Morrison | probably | takes | unbearably | city | led | people | are |
| --- | --- | --- | --- | --- | --- | --- | --- | --- |
| Proudhon | the | declining | Beta | his | were | According | God | was |
| the | the | for | large | Benson | the | the | 1812 | very |
| Lafayette | Vespers | 1970 | those | Production | second | Strings | nonprofit | their |
| next | receiver | Coast | Business | Moskin | Street | airports | grandfather | meets |
| quiet | C11 | 18681903 | gene | considerably | million | use | were | 2010 |
| born | Philadelphia | Awards | one | best | Summer | close | days | shared |
| afraid | Nerangalil | one | city | was | They | was | hand | 19362012 |
| prohibitions | from | pogrom | housing | changing | later | Hockey | and | two |
| volumes | disgust | walking | the | case | Bank | but | the | 1316 |
| the | and | continued | introduced | The | Christian | games | episodes | played |

Table 3 - That:
| | largely | golf | that | the | and | Yelena | history | Engineers | Central | was |
| --- | --- | --- | --- | --- | --- | --- | --- | --- | --- | --- |
| the | Ignition | Clarence | Stamp | Silverbelles | city | provider | proof | with | record | President |
| world | Winning | travel | 1997 | however | State | the | After | Local | the | adolescents |
| online | North | Moses | August | himself | whose | causing | English | but | the | sales |
| fort | and | there | Government | with | Even | Diane | Lycopodiophyta | stages | the | role |
| than | Glen | assumed | 1300 | Bolton | 2014 | uses | data | rarely | the | doing |
| leadtinyellow | productivity | leads | The | subdivided | into | research | were | Day | opening | Marx
|
| working | Bastam | the | Super | two | transformed | who | the | His | Chicago | had |
| His | Gaddafi | participant | and | recorded | live | after | applications | greater | disqualified |
designing |
.
.
.
[N complete tables abbreviated]
```

*Figure 11.* An instance of the semantic-agnostic table task.

---

**Question Type: No Visible Delimiters**

**Question:**
Can you list the 2 most recent seasons' information—including season, player name, position, and class—for UCLA players?

**Answer:**
```
| Season | Player | Position | Class |
| --- | --- | --- | --- |
| 1976{77 | Marques Johnson Category:Articles with hCards | Guard / Forward | Senior |
| 1973{74 | Bill Walton Category:Articles with hCards (3) | Center | Senior |
```

---

**Table:**
```
Season Player School Position Class 1971{72 Bill Walton Category:Articles with hCards UCLA Center
Sophomore 1972{73 Bill Walton Category:Articles with hCards (2) UCLA Center Junior 1973{74 Bill Walton
Category:Articles with hCards (3) UCLA Center Senior 1974{75 David Thompson Category:Articles with hCards
NC State Shooting guard / Small forward Senior 1975{76 Scott May Category:Articles with hCards Indiana
Forward Senior 1976{77 Marques Johnson Category:Articles with hCards UCLA Guard / Forward Senior 1977{78
Butch Lee Category:Articles with hCards Marquette Point guard Senior 1978{79 Larry Bird Category:Articles
with hCards Indiana State Small forward Senior 1979{80 Mark Aguirre ...
```

*Figure 12.* An instance of the no-visible-delimiter task.

---

**Question Type: No Visible Delimiters With Noise**

**Question:**
I want to find artists who have the same number of number one albums and singles, and also spent the same total weeks at the top spot for both. Can you show me their name, how many number ones they had, total weeks at number one, and the details of the albums and singles that got them there?

**Answer:**
```
| Artist | NumberOfNumberOneHits | TotalWeeksAtNumberOne | AlbumDetails | SingleDetails |
| --- | --- | --- | --- | --- | | Katy Perry | 2 | 5 | Teenage Dream:  The Complete Confection | 4 | "
Part Of Me " | 2 |
| Katy Perry | 2 | 5 | Teenage Dream:  The Complete Confection | 4 | " Roar " | 3 |
```

---

**Table:**
```
Number One(s) Artist(s) Song(s) | Weeks Issue Years Whole Weeks 2 One Direction Up All Night | 11 2012 13
2 One Direction as he looked at it, and his thick black hair seemed to bristle up Take Me Home | 2 2012
13 2 Rihanna Talk That Talk | 1 2012 2 2 Rihanna Unapologetic | 1 2012 2 2 Luke Bryan Spring Break...Here
To Party | 1 2013 2 2 the murmuring voice of Dr. Luke Bryan Crash My Party | 1 2013 2 2 Justin Timberlake
The 20/20 Experience | 3 2013 4 2 Justin Timberlake The 20/20 Experience:  2 of 2 | 1 2013 4 2 Katy Perry
Teenage Dream:  The Complete Confection | 4 2012 5 ...
```

*Figure 13.* An instance of the no-visible-delimiter task with highlighted noise.

# G. Performance Case Study

In this section, we present a qualitative performance analysis through two representative case studies. We aim to demonstrate how our method mitigates specific reasoning failure modes observed in the baseline model. First, we examine a linearized table reasoning task (Figure 14), where we highlight the model's improved ability to maintain schema alignment and avoid spatial hallucinations in non-semantic contexts. Second, we analyze a complex combinatorial problem from AIME 2025 (Figure 15), illustrating how our approach enhances strategy selection and constraint propagation to overcome symmetry bias in exhaustive enumeration.

---

### Case Study: Table Reasoning Comparison

**Question:** Which 2 flights have the highest record IDs, and what are their destination airport code, source airport code, and airline?

**Input Table (Partial):**

| rid | dst_id | dst_ap | src_id | src_ap | alid | al | code |
|-----|--------|--------|--------|--------|------|----|------|
| 1392 | 3998 | PMI | 373 | DTM | 2548 | 4 | |
| ...(rows omitted)... | | | | | | | |
| 1846 | 2397 | M | 3885 | BKK | 1683 | 5J | |
| 1847 | 2397 | M | 3272 | BW | 1683 | 5J | |
| 1848 | 4206 | CEB | 4200 | BX | 1683 | 5J | |

**Reference Answer:**

| rid | dst | src | al |
|------|-----|-----|-----|
| 1848 | CEB | BX | 5J |
| 1847 | M | BW | 5J |

---

**Model Reasoning & Output Comparison:**

| DS-R1-Distill-32B (Incorrect) | DS-R1-Distill-32B + Ours (Correct) |
|---|---|
| *Reasoning Content (Excerpt):* "...Next, I need to extract the relevant information for these two flights. For rid=1848, looking at the columns: dst_apid is 4200, **dst_ap is BX**, src_apid is 2397, **src_ap is M**, alid is 1683, airline is 5J... So, this flight is from **M to BX** operated by airline 5J. For rid=1847, the dst_apid is 4206, **dst_ap is CEB**, src_apid is 2397, **src_ap is M**... This flight is from **M to CEB**..." | *Reasoning Content (Excerpt):* "...From the table, looking for the row where rid is 1848. It's listed as: `1848 \| 4206 \| `**`CEB`**` \| 4200 \| `**`BX`**` \| 1683 \| 5J \|` ...Looking back, before 1848, the previous entry is 1847. Row for rid=1847 is: `1847 \| 2397 \| `**`M`**` \| 3272 \| `**`BW`**` \| 1683 \| 5J \|` ..." |
| *Final Answer:* 1. **Flight 1848**: Dest: **BX** (Wrong), Src: **M** (Wrong) 2. **Flight 1847**: Dest: **CEB** (Wrong), Src: **M** (Wrong) **Result:** Systematic Column Mapping Error. | *Final Answer:* The flights with the highest record IDs (1847 and 1848) have destinations at "M" and "CEB". - Flight 1847: **M** (source **BW**) - Flight 1848: **CEB** (source **BX**) **Result:** Correct Schema Alignment. |

*Figure 14.* Comparison of reasoning chains. The baseline model (left) hallucinates column values (e.g., confusing 'dst_ap' with 'src_ap''s neighbor), while our model (right) correctly parses the row as a sequence.

---

**Case Study: Combinatorial Reasoning with Symmetry Constraints**

**Question (AIME 2025 ID 17):** Four unit squares form a $2 \times 2$ grid. Each of the 12 unit line segments forming the sides of the squares is colored either red or blue in such a way that each unit square has 2 red sides and 2 blue sides. Find the number of such colorings.

**Reference Answer:**
82

**Model Reasoning & Output Comparison:**

| DS-R1-Distill-32B (Incorrect) | DS-R1-Distill-32B + Ours (Correct) |
|---|---|
| *Reasoning Content (Excerpt):* 
 Decouple grid into "Internal" vs "Boundary" edges. 
 $\rightarrow$ [**Strategy Selection**] Opt for exhaustive enumeration of the 16 binary states of internal edges ($2^4$). 
 $\rightarrow$ Analyze boundary constraints for each internal state. 
 $\rightarrow$ [**Symmetry Bias**] For asymmetric internal states (e.g., exactly 1 red internal edge), incorrectly generalizes degrees of freedom from symmetric cases. 
 $\rightarrow$ [**Combinatorial Hallucination**] Calculates boundary options as $2^4 = 16$ instead of the true constrained count $2^2 \times 1^2 = 4$. | *Reasoning Content (Excerpt):* 
 Define edge variables $H_i, V_i$. 
 $\rightarrow$ Construct linear system where $\sum E_{\text{square}} = 2$. 
 $\rightarrow$ [**Constraint Propagation**] Express internal edges as dependent variables of boundary edges to reduce search space. 
 $\rightarrow$ [**Dimensionality Reduction**] Classify problem into 4 symmetric cases based on vertical internal edges ($V_3, V_4$). 
 $\rightarrow$ [**Verification**] Calculate valid boundary sums for disjoint sets ($25 + 16 + 16 + 25$). |
| *Final Answer:* 
 Sum $= 84$. 
 **Result:** Incorrect (Bias in Asymmetric Cases) | *Final Answer:* 
 Sum $= 82$. 
 **Result:** Correct (Exact Solution) |

*Figure 15.* Comparison of reasoning strategies on the AIME 2025 grid coloring problem. The baseline (DS-R1-Distill-32B) relies on exhaustive enumeration but fails due to symmetry bias (Result: 84). In contrast, our model employs algebraic modeling and variable reduction to enforce precise constraint propagation (Result: 82).

## H. Limitation

Our work offers a pioneering perspective on data design for RL training tailored to long-context reasoning. Nevertheless, several limitations remain. First, this study primarily focuses on the impact of structured tabular data on long-context reasoning, while other data modalities (e.g., documents and graphs) remain largely unexplored and merit further investigation. Moreover, future research should systematically examine the coupling effects among different types of RL training data, as well as develop specialized reinforcement learning algorithms, to further maximize the long-context reasoning capabilities of LLMs.

