# OpenReview forum: "Probing How Scalable Table Data Enhances General Long-Context Reasoning"
_ICML.cc/2026/Conference — ICML 2026 regular_

### Official Review · Reviewer_nCYD · 2026-03-12

**Soundness:** 3
**Presentation:** 3
**Significance:** 2
**Originality:** 3
**Overall Recommendation:** 4
**Confidence:** 4

**Summary:**

This paper proposes TableLong, an automated pipeline that uses SQL execution to synthesize scalable, verifiable tabular data and QA tasks for LLM reinforcement learning. The core premise is that structured tabular data inherently contains periodic, non-vanishing dependencies that force models to maintain attention across extended contexts. The authors support this with an information-theoretic analysis based on mutual information. Empirical results on DeepSeek-R1-Distill-Qwen models indicate that RL training on these synthetic tables improves performance on standard long-context benchmarks and shows positive transfer to out-of-domain tasks such as math and coding.

**Compliance With Llm Reviewing Policy:**

Affirmed.

**Key Questions For Authors:**

1. Do you have any preliminary results applying TableLong to larger models (e.g., 32B+)? Does the out-of-domain generalization to math and coding hold as model scale increases?

2. Given the theoretical focus on periodic dependencies, is there a risk of the model overfitting to these rigid structures? Did you evaluate performance on long-context tasks characterized by highly irregular or non-periodic dependencies?

3. How does the pipeline ensure sufficient diversity in the SQL-generated queries? Is it possible that the RL policy is simply learning to internally translate natural language to SQL, rather than fundamentally improving its general reasoning capacity?

**Limitations:**

The authors address limitations in Section H, noting their exclusive focus on tabular data and acknowledging the need to explore other modalities (e.g., long documents, graphs) and the coupling effects between mixed RL training data types.

**Strengths And Weaknesses:**

Soundness

Strengths: The theoretical framing is a strong point. Using mutual information to quantify the periodic dependencies in tabular data provides a solid mathematical justification for the method, moving beyond pure empirical heuristics. Furthermore, relying on SQL execution to provide deterministic, verifiable ground-truth rewards is a robust design choice that effectively mitigates reward hacking during RL.

Weaknesses: The empirical validation is currently limited to smaller distilled models (1.5B and 7B parameters). While these experiments successfully demonstrate the trend, testing on larger-scale models (e.g., 32B or 70B) is necessary to definitively prove that this synthetic data approach scales to frontier-level capabilities.

Presentation

Strengths: The manuscript is clearly written and logically structured. The narrative smoothly transitions from the information-theoretic analysis to the pipeline design and empirical results. Figures and tables are effectively utilized to show scaling behaviors.

Weaknesses: The main text relies too heavily on the appendices for examples of the generated TableLong queries. Including a few concrete, representative examples directly in the methodology section would give readers a better immediate grasp of the task complexity and the exact nature of the synthetic data.

Significance

Strengths: Constructing high-quality, verifiable data for long-context RL remains a critical bottleneck. Showing that synthesized tabular data can act as an effective proxy for general long-context reasoning offers a practical, scalable recipe for post-training. The observed out-of-domain generalization to math (AIME) and coding tasks is an especially compelling result.

Weaknesses: The transferability to out-of-domain tasks, while impressive, lacks a deep mechanistic explanation. It is not entirely clear why tabular reasoning specifically transfers to AIME or coding, as opposed to simply serving as a rigorous cognitive exercise.

Originality

Strengths: While training LLMs on tabular data is common, reframing it as a vehicle specifically for long-context RL—backed by the mutual information analysis of periodic dependencies—is an original perspective. The TableLong pipeline creatively combines SQL-based synthesis with exact reward verification, offering a clean solution to reward hallucination.

Weaknesses: The novelty lies more in the application and theoretical justification of the data rather than in the RL pipeline itself. The underlying RL algorithms and the concept of using verifiable environments (like SQL/code execution) for rewards are well-established. The contribution is heavily data-centric rather than algorithmic.

---

> ### Author Rebuttal · Authors · 2026-03-31
>
> **We sincerely thank the reviewer for the thorough evaluation and for recognizing our theoretical framing, verifiable reward design, and OOD generalization results. We address each concern below.**
>
> ---
>
> ### **[Weakness 1 & Question 1] Model Scale of Empirical Validation**
>
> We respectfully clarify a misunderstanding: our experiments are conducted on **14B and 32B models** (DS-R1-Distill-Qwen-14B/32B, Qwen2.5-32B-Instruct, Qwen3-32B), not 1.5B or 7B as the review suggests. As shown in Table 1:
>
> | Model | Baseline Avg. | + Ours | Gain |
> |-------|--------------|--------|------|
> | DS-R1-Distill-14B | 36.14 | 45.07 | **+8.93%** |
> | DS-R1-Distill-32B | 40.12 | 48.36 | **+8.24%** |
> | Qwen2.5-32B-Instruct | 36.15 | 40.94 | **+4.79%**|
> | Qwen3-32B | 40.40 | 45.34 | **+4.94%** |
>
> These results demonstrate the robustness of TableLong across different backbone types and scales. Furthermore, the improvements generalize well to OOD domains, with gains of +10.00% on AIME 2025 and +11.97% on LiveCodeBench for DS-R1-Distill-32B (Table 4).
>
> ---
>
> ### **[Weakness 2] Appendix Reliance for Data Examples**
>
> We appreciate this suggestion. The representative examples of generated TableLong queries were placed in the appendix primarily due to page constraints. In the revised version, we will consider incorporating a few concise, representative examples directly into the methodology section to give readers a more immediate grasp of the task complexity and the nature of the synthetic data.
>
> ---
>
> ### **[Weakness 3] Mechanistic Explanation for OOD Transfer**
>
> While our primary scope is to probe the mechanism by which table data enhances long-context reasoning, the OOD transfer is not coincidental. Long-context reasoning and domain-specific capabilities (math, code) are largely **orthogonal dimensions**; the gains in both arise from the same core skills TableLong cultivates: **precise retrieval**, **multi-hop reasoning**, and **grounding**. These are domain-agnostic abilities — multi-step derivation in AIME and logic tracing in code demand the same long-range attention and structured reasoning that table training strengthens. This is empirically confirmed by Table 4 (+10.00% on AIME 2025, +11.97% on LiveCodeBench).
>
> ---
>
> ### **[Weakness 4] Data-Centric vs. Algorithmic Contribution**
>
> We respectfully note that data-centric contributions are widely recognized as first-class contributions in the LLM community. The choice of training data fundamentally shapes both training stability and performance ceiling — as demonstrated by influential works such as LoongRL [1] and OpenCoder [2], which advanced their respective fields primarily through data-centric insights rather than algorithmic novelty.
>
> As stated in our title, the core contribution of this paper is to *probe the intrinsic mechanism by which scalable table data enhances general long-context reasoning*, supported by both a principled mutual information analysis and comprehensive empirical investigation. We believe understanding *why* a data type works is itself a valuable and original contribution.
>
> ---
>
> ### **[Question 2] Risk of Overfitting to Periodic Structures**
>
> Overfitting to periodic structures is not a concern: all our evaluation benchmarks are non-tabular, and TableLong achieves strong improvements across all of them (Table 1). Two benchmarks are especially relevant as they feature highly irregular, non-periodic dependencies:
>
> - **LongBench-v2**: Real-world multi-task benchmark with entirely unstructured, non-periodic text.
> - **MRCR**: Multi-turn conversational retrieval with randomly distributed information across long contexts.
>
> Consistent improvements on these benchmarks confirm that the model generalizes well beyond periodic table structures.
>
> ---
>
> ### **[Question 3] Query Diversity and Reasoning Mechanism**
>
> Diversity is ensured through a structured constraint system spanning three task dimensions — **Precise Retrieval**, **Multi-hop Reasoning**, and **Grounding**. For each dimension, a pre-defined candidate pool of logical instructions (e.g., targeting key, categorical, or numerical columns) is randomly sampled to produce SQL queries with varied conditions and structures, ensuring high diversity and quality through deterministic rules rather than unconstrained LLM generation.
>
> Regarding NL-to-SQL learning: SQL is solely an intermediate tool for data construction — it is never exposed to the model during RL, and no SQL execution tool is provided. As shown in Figure 14, the model's CoT consists entirely of multi-step natural language reasoning with no SQL generation, confirming that it learns structured long-range reasoning rather than query translation.
>
> ---
>
> ### Reference
>
> [1] Wang, S., et al. "LoongRL: Reinforcement Learning for Advanced Reasoning over Long Contexts." 2025.
>
> [2] Huang, S., et al. "OpenCoder: The Open Cookbook for Top-Tier Code Large Language Models." 2024.
>
> ---
>
> **We hope our response addresses your concerns and welcome any further feedback.**

---

### Official Review · Reviewer_hgZL · 2026-03-12

**Soundness:** 3
**Presentation:** 3
**Significance:** 3
**Originality:** 2
**Overall Recommendation:** 4
**Confidence:** 3

**Summary:**

This paper investigates how structured table data enhances the long-context reasoning capabilities of large language models (LLMs). The authors demonstrate that table data contains periodic, non-vanishing dependencies, which are mathematically verified through mutual information analysis. They propose a scalable pipeline called TableLong for synthesizing high-quality, diverse, and verifiable structured table data, which, when used in reinforcement learning training, significantly improves LLM performance on various long-context and out-of-domain benchmarks.

**Compliance With Llm Reviewing Policy:**

Affirmed.

**Final Justification:**

I maintain my original score.

**Key Questions For Authors:**

N/A

**Limitations:**

yes

**Strengths And Weaknesses:**

Strengths
- The proposed TableLong pipeline is scalable, flexible, and capable of generating diverse, high-quality, verifiable structured data tailored for long-context tasks.
- Extensive experiments across multiple benchmarks show that training with table data leads to significant improvements in reasoning, retrieval, and out-of-domain generalization.

Weaknesses
- In terms of originality, the cells in the same column of the table data come from the same distribution, so there is still a dependency relationship between the cells in the same column after linearization. That is, the periodic non-vanishing structure of the table data is quite evident. This weakens the significance of the proof presented in the article.
- Regarding rationality, firstly, how to ensure the correctness of synthetic data, that is, the generated SQL can correctly answer the generated questions. Secondly, can the author provide the results of directly using these data for SFT? Additionally, what model did the author use to synthesize data, and has the author tried using different models to synthesize data to compare their effects.

---

> ### Author Rebuttal · Authors · 2026-03-31
>
> **We thanks for the careful reading and recognition of our work. We address each concern below.**
>
> ---
>
> ### **[Question 1] Theoretical Significance**
>
> We acknowledge that column-wise distributional consistency (Assumption A1) is intuitive, but our contribution extends **significantly beyond** this observation.
>
> #### **(1) Precise quantitative characterization, not mere existence of dependency.**
>
> Our contribution is **not** restating that "same-column cells share a distribution." We provide an information-theoretic framework with several precise, non-trivial results:
>
> - **Exact quantification at periodic lags** (Theorem 2.2): The average MI at every lag $d = km$ equals exactly $I_{\text{same}}$, independent of $k$—establishing that dependency strength is **perfectly constant** across arbitrarily large distances.
>
> - **Row-distance invariance** (Lemma A.8(ii)): $I(T_{i,J}; T_{k,J}) = I_{\text{same}}$ for **any** $i \neq k$—MI between same-column cells does not decay with row separation, a non-trivial result requiring formal proof.
>
> - **Characterization of $I_{\text{same}}$ magnitude** (Proposition A.11): $I_{\text{same}}$ is governed by cross-column variance $$\sigma^2 = \sum_a \text{Var}_j[P_j(a)]$$ precisely identifying **what determines** long-range dependency strength.
>
> #### **(2) The core significance is the systematic contrast with natural language.**
>
> Our primary contribution is a rigorous **comparative framework** between table data and natural language:
>
> | Property | Natural Language | Table Data |
> |---|---|---|
> | MI decay | $I_{\text{text}}(d) \sim C \cdot d^{-\alpha} \to 0$ (Eq. 1) | $\bar I_{\text{table}}(km) = I_{\text{same}} > 0$ (Theorem 2.2) |
> | Asymptotic behavior | $\lim_{d \to \infty} I_{\text{text}}(d) = 0$ | $\liminf_{d \to \infty} \bar I_{\text{table}}(d) \geq I_{\text{same}} > 0$ (Corollary 2.3) |
> | Asymptotic dominance | — | $\lim_{k \to \infty} \frac{\bar I_{\text{table}}(km)}{I_{\text{text}}(km)} = +\infty$ (Corollary 2.4) |
> | Effective dependency distance | $D_{\text{eff}}^{\text{text}}(\tau) < \infty$ | $D_{\text{eff}}^{\text{table}}(\tau) = +\infty$ (Theorem 2.6) |
>
> These results answer: **"Why is table data more suitable than natural text for long-context reasoning training?"** Corollary 2.4 and Theorem 2.6 are **not** deducible from the starting observation—they require formally connecting the periodic structure to MI dynamics against the power-law decay model (Li, 1989; Lin & Tegmark, 2016).
>
> #### **(3) Formalizing intuitions is a well-recognized contribution in information theory.**
>
> Classic results such as **Shannon's source coding theorem** and **the data processing inequality** formalize intuitive observations into rigorous frameworks, yet are among the most impactful results in the field. More recently, **L²M (Chen et al., NeurIPS 2025)** formalizes long-range dependencies in natural language via MI scaling laws. Our work is, **to our knowledge, the first** such analysis for tabular data—filling a gap in the literature and providing a principled foundation for understanding why table data is effective for long-context reasoning.
>
> #### **(4) Actionable, non-obvious guidance for data construction.**
>
> Proposition A.11 yields concrete guidance: $I_{\text{same}}$ increases monotonically with $\sigma^2$, so prioritizing tables with **diverse column types** (numerical, categorical, temporal, textual) maximizes training effectiveness. Remark A.12 further suggests columns with disjoint value spaces yield the strongest $I_{\text{same}}$.
>
> ---
>
> ### **[Question 2] Data Correctness, SFT Comparison, and Synthesis Model**
>
> **Data correctness** is ensured at two levels. First, SQL execution against the database guarantees the logical consistency of each (Raw Table, Question, Answer) triplet. Second, the consistency-based filtration discards samples with $P=0$ (ambiguous, erroneous, or beyond reasoning capacity), **further eliminating potentially erroneous instances**. Regarding the synthesis model, we use **DeepSeek-v3.1**, which we find sufficiently capable for this task. We plan to explore the impact of different synthesis models in future work.
>
> For the SFT comparison, we use the best response from 16 RL rollouts per instance as the SFT target and fine-tune the same backbone under identical settings:
>
> | Method | LB-v2 | Loong | BrowsComp | Oolong-Synth | MRCR | Mean |
> |--------|-------|-------|-----------|--------------|------|------|
> | SFT | 42.55 | 38.89 | 68.20 | 41.03 | 33.47 | 44.83 |
> | RL | 45.73 | 45.30 | 74.31 | 51.41 | 40.57 | **51.46** |
>
> RL consistently outperforms SFT across all benchmarks (+6.63% on average), demonstrating that reward-driven exploration is essential for fully unlocking long-context reasoning from table data.
>
> ---
>
> **We hope our response addresses your concerns, and welcome any further questions or suggestions.**

---

> > ### Author Rebuttal · Reviewer_hgZL · 2026-04-02
> >
> > Thank you for the author's reply. I think the dependencies of cells in the table can be intuitively explained. I will maintain my original score

---

> > > ### Author Response · Authors · 2026-04-07
> > >
> > > Thank you for your follow-up. We respect your decision, but would like to maintain our position for the record.
> > >
> > > We agree that the existence of same-column dependency is intuitive. However, our contribution is precisely what intuition cannot provide:
> > >
> > > Exact quantification (Thm. 2.2, Lemma A.8): MI equals exactly $I_{\text{same}}$ at every periodic lag, invariant to row distance—not deducible from intuition alone.
> > > Asymptotic dominance over natural language (Cor. 2.4, Thm. 2.6): $\bar I_{\text{table}}$ asymptotically dominates $I_{\text{text}}$, and $D_{\text{eff}}^{\text{table}}(\tau) = +\infty$ vs. $D_{\text{eff}}^{\text{text}}(\tau) < \infty$—results that require formally connecting periodic structure to the power-law decay model of natural language.
> > > Actionable guidance (Prop. A.11): $I_{\text{same}}$ is governed by cross-column variance $\sigma^2$, yielding concrete prescriptions for data construction.
> > > Formalizing intuitions into rigorous, quantitative frameworks is a well-recognized contribution in information theory. We respectfully disagree that intuitiveness diminishes the value of such characterization.

---

### Official Review · Reviewer_wPQH · 2026-03-13

**Soundness:** 3
**Presentation:** 3
**Significance:** 3
**Originality:** 3
**Overall Recommendation:** 5
**Confidence:** 3

**Summary:**

This paper shows that tabular data is well-fitted for long-context reasoning. It starts with theoretical explanations on the periodic non-vanishing dependencies present in tabular data. It then shows that a careful synthesis of high-quality tabular data can boost the long-context reasoning ability of LLMs, even in the OOD setting.

**Compliance With Llm Reviewing Policy:**

Affirmed.

**Final Justification:**

The authors have addressed my concerns.

**Key Questions For Authors:**

1. In Section 3.4, you mentioned applying a dual-sided filter to preserve non-trivial tasks. Do you have any intuition on what will happen if, say, we retain tasks with $0.2 < P < 0.8$? In other words, do you think the current filtering of $0 < P < 1$ is optimal?

2. Do you have any intuition on what kinds of real-world tables are the most useful for enhancing long-context reasoning? Does the answer change if, say, we are particularly interested in improving long-context reasoning for a specific domain (e.g., retrieval)?

**Limitations:**

Yes

**Strengths And Weaknesses:**

**Soundness:** This paper is moderately sound. Theoretical explanations are provided, and experiments clearly show the benefit of the proposed approach.

**Presentation:** This paper is well-written and easy to follow.

**Significance:** This paper addresses an important problem of long-context reasoning and the result shows the potential of this method on improving it.

**Originality:** This paper builds on existing works on the use of tabular data for long-context reasoning, but provides theoretical explanations  and suggests effective ways of constructing tabular data to improve long-context reasoning.

---

> ### Author Rebuttal · Authors · 2026-03-31
>
> **We sincerely thank the reviewer for the thoughtful and constructive feedback, and for the positive assessment of our theoretical analysis, pipeline design, and experimental results. We address each question below.**
>
> ---
>
> ### **[Question 1] Filtering Threshold and Optimality**
>
> The goal of our $0 < P < 1$ filter is to ensure task quality and non-triviality: tasks with $P=0$ are typically ill-posed, erroneous, or beyond the model's current reasoning capacity, while $P=1$ tasks are already fully mastered and thus trivially solvable, providing no meaningful learning signal.
>
> Your question raises an important follow-up. A stricter range such as $0.2 < P < 0.8$ could potentially improve sample efficiency, as medium-difficulty tasks yield more stable advantage signals and higher rollout variance, leading to more effective gradient updates.
>
> That said, the primary scope of this work is to probe the fundamental mechanism by which table data enhances long-context reasoning, rather than to optimize every pipeline component. The current filter is designed to be principled and broadly applicable across different settings.
>
> We greatly appreciate your insightful suggestion and plan to explore the effect of different pass-rate intervals on training efficiency in future work.
>
> ---
>
> ### **[Question 2] Table Type Selection and Domain Adaptation**
>
> Based on our experience, data-dense real-world tables — such as financial reports, property listings, and scientific records — tend to be the most beneficial for long-context reasoning. These tables naturally require the model to perform cross-column computation and multi-row aggregation, while producing precise and verifiable answers that yield stable reward signals during RL training.
>
> For retrieval-specific domains, our findings suggest that TableLong already provides substantial benefits. As shown in Figure 3, models trained with table data exhibit significantly stronger grounding ability, **achieving notable NIAH improvements across different backbones (+21.90% for the 14B model, +11.45% for the 32B model)**. If one is particularly interested in further improving retrieval performance, **we recommend prioritizing multi-hop queries, cross-table JOIN operations, and knowledge-sparse tasks during data construction**, as these task types more directly challenge the model's ability to locate and associate distributed information across long contexts.
>
> ---
>
> **We hope that our response has adequately addressed your concerns. We sincerely welcome any further questions or suggestions and greatly appreciate your feedback in helping us improve the paper.**

---

> > ### Author Rebuttal · Reviewer_wPQH · 2026-04-03
> >
> > Thanks for the response. I am raising my score to 5.

---

> > > ### Author Response · Authors · 2026-04-07
> > >
> > > We thank the reviewer for the positive feedback and for increasing the score. We are very encouraged that our response has addressed your concerns and that you find the paper's contributions clearer.

---

### Official Review · Reviewer_SzwM · 2026-03-14

**Soundness:** 3
**Presentation:** 3
**Significance:** 3
**Originality:** 3
**Overall Recommendation:** 4
**Confidence:** 4

**Summary:**

This paper investigates why structured table data is effective for improving long-context reasoning in LLMs. The key theoretical insight is that when tables are linearized, they exhibit periodic non-vanishing mutual information. Based on this analysis, the authors propose TableLong, a pipeline that synthesizes SQL-based table tasks (retrieval, multi-hop reasoning, grounding) with verifiable answers, filters them via a dual-sided consistency check, and uses them for RL training (GRPO).

**Compliance With Llm Reviewing Policy:**

Affirmed.

**Final Justification:**

Rebuttal addressed my questions.

**Key Questions For Authors:**

Q1. What if train the model with verifiable non-tabular long-context data such as multi-file code tasks. This could isolate whether the gain from table structure or verifiable long context RL.
Q2. What's the filtering rate of dual-sided filtering process, i.e. P=0 and P=1. If the retention is low, how could the pipeline scale up?

**Strengths And Weaknesses:**

The strength of the paper
S1: The mutual information analysis of linearized tables is clean. The periodic non-vanishing dependencies in tables is a novel observation the provides a principled explanation for why table data helps with long context reasoning.
S2: The proposed solution is practical and reproducible. The TableLong pipeline is straightforward. The data sources are public and training details are comprehensive.
S3: The experiments with ablations are thorough and interesting. The evaluation is comprehensive, covering 7 long context benchmarks and 4 OOD benchmarks, with scaling experiments across length, cell count and table count.

The weakness of the paper:
W1: The theoretical analysis assume ideal table setting. In practice, tables could have missing values, hierarchical headers, sparse tables. Whether the theoretical conclusion still holds with all these variations?
W2. The paper claims "scalable" table data, but the training is only done at 4k, 8k, 16k, not very long. Any case that go beyond 16k? Also, how this synthetic data based on tables compared with other approaches, like coding problem, long-form math problems?
W3. The NIAH results need deeper explanation. the baseline for a 14B model is abnormally low. The RL training may be recovering lost capability rather than improved capability from structural properties. Could the author also report NIAH for the base model before distillation to distinguish recovery from genuine improvement?

---

> ### Author Rebuttal · Authors · 2026-03-31
>
> **We thanks for the careful reading and constructive feedback, and address each concern below.**
>
> ---
>
> ### **[Weakness 1] Theory Robustness**
>
> Our framework targets the **fundamental mechanism**—the periodic non-vanishing dependency structure. As noted in Remark A.7, we do **not** assume conditional independence within columns, so results extend to sorted data and time series. Real-world variations (missing values, sparsity, hierarchical headers) may **attenuate** the periodic MI peaks but do not eliminate them, introducing only **quantitative** perturbations without **qualitatively** altering the contrast with natural language (Corollary 2.3, Theorem 2.6).
>
> Section 4.3.1 directly demonstrates this robustness:
>
>
> | Setting | Description | Avg. Accuracy | Δ |
> |---|---|---|---|
> | **Ours (full)** | Complete table with all structures | 48.36% | — |
> | **No visible delimiters** | All formatting delimiters removed | 47.59% | -0.77% |
> | **No visible delimiters + noise** | Delimiters replaced with random noise text | 47.38% | -0.98% |
> | **Baseline** | Baseline | 40.12% | — |
> | **No semantics** | All cell contents replaced with random words | 41.79% | +1.67% over baseline |
>
>
> The most aggressive perturbation—**"no visible delimiters + noise"** (Figure 13)-caused only a **0.98%** drop, with reward curves confirming convergence to a comparable asymptotic level (Figure 4b). Even the **"no semantics"** setting improves over baseline by **+1.67%**, showing the periodic structure alone provides value. Furthermore, training on real-world BIRD/CoSQL/Spider data (Section 3.2) yields **+8.24%** across seven OOD benchmarks (Table 1), confirming practical effectiveness.
>
> ---
>
> ### **[Weakness 2 & Question 1] Scalability and Structural Advantage**
>
> **On length scalability beyond 16k (W2).**
> Our main experiments were conducted at 4k, 8k, and 16k due to computational constraints; Tables 9 and 10 already show strong extrapolation to 128k. To further address this concern, we provide a supplementary 32k training experiment on DS-R1-Distill-32B:
>
>
> | Method | LB-v2 | Loong | BrowsComp | Oolong-Synth | MRCR | Mean |
> |--------|-------|-------|-----------|--------------|------|------|
> | Ours (16k) | 45.73 | 45.30 | 74.31 | 51.41 | 40.57 | 51.46 |
> | Ours (32k) | 47.72 | 46.90 | 76.41 | 51.90 | 44.24 | **53.43** |
>
>
> Scaling to 32k yields a consistent **+1.97%** average improvement, confirming that TableLong scales effectively beyond 16k.
>
>
>
> **On the advantage of table data over other verifiable data types (W2 & Q1).**
> As established in Theorem 2.2 and Corollary 2.3, table data possesses **periodic non-vanishing structural dependencies** absent in other modalities (e.g., documents, math), making it uniquely suited for long-context reasoning training.
>
> To empirically validate this, we compare against **LoongRL**, a verifiable non-tabular long-context RL dataset built on unstructured text with rule-based answers, replacing only the training data while keeping all other settings identical—isolating the effect of *table structure* from *verifiable long-context RL* alone.
>
>
> | Method | LB-v2 | Loong | BrowsComp | Oolong-Synth | MRCR | Mean |
> |--------|-------|-------|-----------|--------------|------|------|
> | LoongRL | 45.73 | 40.29 | 70.74 | 43.84 | 37.23 | 47.57 |
> | Ours (16k) | 45.73 | 45.30 | 74.31 | 51.41 | 40.57 | **51.46** |
>
>
> TableLong outperforms LoongRL by +3.89% on average, confirming that the gains stem substantially from the intrinsic structural properties of table data, beyond verifiable long-context RL alone.
>
> ---
>
> ### **[Weakness 3] NIAH Baseline Analysis**
>
> We appreciate this insightful observation. The lower NIAH baseline of DS-R1-Distill-14B (69.30%) is indeed consistent with the known phenomenon that knowledge distillation can cause certain capability degradation. To provide a more complete picture, we additionally report NIAH results on the non-distilled **Instruct** backbones:
>
>
> | Model | NIAH (Base) | NIAH (+ TableLong) | Δ |
> |-------|------------|-------------------|---|
> | Qwen2.5-14B-Instruct | 94.50% | 96.55% | +2.05% |
> | DS-R1-Distill-14B | 69.30% | 91.20% | +21.90% |
> | Qwen2.5-32B-Instruct | 97.55% | 98.55% | +1.00% |
> | DS-R1-Distill-32B | 87.95% | 99.40% | +11.45% |
>
>
> Despite strong NIAH baselines (94.50% and 97.55%), the Instruct models still gain +2.05% and +1.00% from TableLong, demonstrating broad effectiveness across backbone types.
>
> ---
>
> ### **[Question 2] Filtering Rate and Pipeline Scalability**
>
> Our candidate pool contains $\sim$110k samples. The dual-sided filter discards $\sim$14% at P=0 (ambiguous/unsolvable) and $\sim$27% at P=1 (already mastered), retaining $\sim$59% ($\sim$65k) for RL training—sufficient in practice.
>
> Scalability is not a concern: since SQL queries and NL questions are synthetically generated, the pool can be expanded by collecting more real-world tables, which are abundantly available.
>
> ---
>
> **We hope that our response has adequately addressed your concerns.**

---

> > ### Author Rebuttal · Reviewer_SzwM · 2026-04-03
> >
> > Thank you for your response, my questions have been resolved.

---

> > > ### Author Response · Authors · 2026-04-07
> > >
> > > We thank the reviewer for the constructive comments throughout the process and for the positive re-evaluation of our work. We are pleased to have resolved all the concerns.

---

### Decision · Program_Chairs · 2026-04-30

**Decision:**

Accept (regular)

**Comment:**

This paper investigates why structured table data enhances long-context reasoning in LLMs. It provides a mutual information analysis of periodic non-vanishing dependencies and proposes the TableLong pipeline for synthesizing verifiable tabular RL training data.

Reviewers gave scores of 5/4/4/4 and recognized the clean theoretical framing and strong empirical results. One reviewer questioned the novelty of formalizing intuitively obvious table dependencies; the authors provided a detailed rebuttal distinguishing precise quantitative characterization from intuitive observation. Other concerns regarding scalability were satisfactorily resolved.

I recommend acceptance. The camera-ready should include representative data examples in the main text rather than relying on appendices.